# Enhanced hybridization-proximity labeling discovers protein interactomes of single RNA molecules

Karen Yap [1,5], Tek Hong Chung [1,5], Erin C. Hedges[2,3], Agnes L. Nishimura[2,3,4], Christopher E. Shaw[2,3] & Eugene V. Makeyev [1] ✉

RNAs engage diverse protein partners and localize to specific subcellular compartments, yet dissecting proteomes associated with low-abundance or dispersed RNA molecules remains a challenge. We present an enhanced hybridization-proximity labeling (HyPro) technology for in situ proteome profiling of endogenously expressed RNA microcompartments. We re-engineer the HyPro enzyme and optimize proximity biotinylation conditions to identify proteins associated with compact RNA-containing nuclear bodies, small pre-mRNA clusters, and individual transcripts. Applying this approach to pathogenic G4C2 repeat–containing C9orf72 RNAs, retained as single-molecule foci in the nuclei of amyotrophic lateral sclerosis (ALS) patient-derived pluripotent stem cells, we reveal extensive interactions with disease-linked paraspeckle markers and a specific set of pre-mRNA splicing factors. These findings highlight early RNA processing and localization defects in ALS that may contribute to this late-onset neurodegenerative disorder. Overall, HyPro provides a broadly applicable platform for mapping RNA-protein interactions, enabling insights into RNA biology and its dysregulation in disease.

RNA transcripts exist in crowded intracellular environments, engaging in interactions with diverse protein ligands[1–5]. These molecular contacts are dynamically remodeled as RNAs are processed or transported to different cellular locations[6–8]. Disease-associated mutations can disrupt the RNA-protein interactome, altering RNA processing, localization, and stability[9–12].

A prominent example of such deregulation is the genetic expansion of the simple GGGGCC (G4C2) repeat in the *C9orf72* gene, which underlies many cases (C9) of amyotrophic lateral sclerosis (ALS) and frontotemporal dementia (FTD) – neurodegenerative disorders characterized by the progressive loss of motor and cortical neurons, respectively[13–15]. This mutation gives rise to aberrant repeat-containing transcripts that tend to accumulate in the nucleus, either as individual

RNA molecules or in small clusters[16–18]. Earlier studies have used recombinant G4C2-containing transcripts to model *C9orf72*-linked pathologies[19–22]. However, a systematic analysis of proteins associated with endogenous C9orf72 transcripts in C9-ALS/FTD patient cells – which could refine our understanding of the molecular mechanisms driving neurodegeneration – has been hindered by the lack of proteomics approaches applicable to low-abundance RNA targets in genetically unperturbed cells.

RNA-centric protein interactome discovery methods, such as those utilizing antisense oligonucleotide pull-down or RNase H enrichment of cross-linked endogenous ribonucleoprotein complexes, have been successfully applied to highly abundant RNAs, including the pre-ribosomal RNA 47S/45S, small nuclear RNA (snRNA)

[1]Centre for Developmental Neurobiology, King's College London, London, UK. [2]Maurice Wohl Clinical Neuroscience Institute, Institute of Psychiatry, Psychology and Neuroscience, King's College London, London, UK. [3]UK Dementia Research Institute Centre, Institute of Psychiatry, Psychology and Neuroscience, King's College London, London, UK. [4]Blizard Institute, Barts and The London School of Medicine and Dentistry, Queen Mary University of London, London, UK. [5]These authors contributed equally: Karen Yap, Tek Hong Chung. ✉e-mail: eugene.makeyev@kcl.ac.uk

U1, and the long non-coding RNAs (lncRNAs) XIST and NORAD[23–26]. Comparable sensitivity has been achieved using hybridization-based proximity labeling approaches, which enable the discovery of proteins neighboring unmodified RNA molecules of interest. One such method, O-MAP, has been recently used to analyze the proteomes associated with 47S/45S, snRNA 7SK, and lncRNA XIST[27]. Similarly, we have developed a hybridization-proximity labeling (HyPro) technology, applying it to 47S/45S, the well-expressed lncRNA NEAT1, and the cancer-specific perinucleolar compartment (PNC)[28,29]. The PNC, the smallest RNA-based entity analyzed by the HyPro-based proteomics (HyPro-MS) to date, still contains several dozen copies of the lncRNA PNCTR[28,29].

The ability to dissect protein interactomes of substantially smaller RNA-containing compartments could provide insights into C9-ALS/FTD mechanisms and facilitate a broader range of studies, including those focused on the mechanisms of transcription site-proximal RNA processing[30–32]. Interestingly, newly synthesized protein-coding transcripts have been shown to transiently accumulate near their transcription sites, where they can complete splicing before being exported from the nucleus to the cytoplasm as mature mRNAs[33]. The steady-state number of pre-mRNA molecules localized near transcription sites likely depends on promoter activity, RNA processing efficiency, and diffusion rates. However, published RNA-FISH analyses indicate that even for actively transcribed genes, such as *ACTB* (which encodes β-actin), the number of pre-mRNA molecules near a transcription site may be considerably smaller than the number of PNCTR molecules in the PNC[34]. Analyzing the proteomes of such ribonucleoprotein (RNP) microcompartments – which we define as those containing <10 RNA molecules – remains a major technical challenge.

With this in mind, we have developed an enhanced version of the HyPro technology by redesigning its key reagent, the recombinant HyPro enzyme, and introducing critical changes to the labeling procedure. We demonstrate that the improved method can refine our understanding of the perinucleolar compartment and provide insights into substantially smaller RNA-containing structures, such as the *ACTB* transcription site and endogenously expressed individual C9orf72 transcripts containing pathologically expanded G4C2 hexanucleotide repeats.

## Results

### Modified HyPro enzyme improves labeling of RNA-containing microcompartments

HyPro technology involves recruiting a proximity biotinylation domain (APEX2 derivative of APX peroxidase) to an RNA target of interest in fixed and permeabilized cells via a digoxigenin (DIG)-binding domain, which interacts with DIG-modified antisense oligonucleotide probes[28,35,36] (Fig. 1A). We reasoned that adapting this proximity labeling approach to RNA microcompartments would require enhancing the activity of the two-domain HyPro enzyme to ensure efficient labeling with a limited number of DIG-modified, target-specific oligonucleotides.

It has been noted that the K14D and E112K mutations introduced into the wild-type soybean ascorbate peroxidase (APX) sequence to generate its APEX and APEX2 derivatives – optimized for monomeric behavior – may reduce enzymatic activity due to less efficient heme incorporation[35,37]. Since HyPro labeling is performed in vitro using a diluted solution of the HyPro enzyme, we hypothesized that reverting these mutations might enhance its activity without promoting multimerization. Introducing the D14K and K112E mutations and removing the N-terminal T7 tag present in the original construct (Fig. 1A) did not affect protein expression or solubility (Supplementary Fig. 1A). Size-exclusion chromatography showed that the modified enzyme was in fact, less prone to multimerization in vitro compared to its original counterpart (Supplementary Fig. 1B, C).

Notably, the modified enzyme exhibited consistently higher peroxidase activity than the original when tested at identical concentrations in solution (Fig. 1B). This improvement was also observed in peroxidase assays where the enzyme was recruited to immobilized DIG-modified probes through its DIG-binding domain (Fig. 1C). We refer to the modified enzyme as HyPro2 throughout this study.

To test whether HyPro2 improves proximity biotinylation of RNA microcompartments, we turned to the *ACTB* transcription site. We first estimated the number of ACTB transcripts associated with this compartment in HeLa cells using single-molecule RNA fluorescence in situ hybridization (RNA-FISH). ACTB-specific probe sets were designed to detect mature mRNAs and full-length or nearly full-length pre-mRNAs (24 antisense oligonucleotides annealing to the last and penultimate exons; ACTB-Ex), as well as all pre-mRNA transcripts (34 antisense oligonucleotides annealing to all introns; ACTB-Int) (Supplementary Fig. 2A and Supplementary Data 1).

The ACTB-Ex probe set identified numerous ACTB mRNA spots in the cytoplasm, which were typically negative for ACTB-Int staining (Supplementary Fig. 2A). In the nucleus, the ACTB-Ex probes revealed 2-3 RNA foci colocalizing with the ACTB-Int probes (Supplementary Fig. 2A). Given that HeLa cells tend to contain three complements of chromosome 7[38], which encodes *ACTB*, these nuclear foci likely correspond to clusters of newly synthesized RNAs near *ACTB* transcription sites. Using the mRNA spots (arrow in Supplementary Fig. 2A close-up) as single-molecule intensity standards, FISH-quant (v2)[39] analysis estimated a median of only 6 pre-mRNA molecules per *ACTB* transcription site.

As a control, we stained HeLa cells with non-repetitive and UC-repeat-specific probes against the lncRNA PNCTR (PNCTR-NR and PNCTR-UC, respectively; Supplementary Fig. 2B and Supplementary Data 1). The PNCTR-containing perinucleolar compartments (PNCs) were visibly larger than the nuclear ACTB foci (Supplementary Fig. 2B). FISH-quant analysis of these compartments, along with diffraction-limited spots occasionally observed nearby and likely corresponding to individual PNCTR molecules (arrow in Supplementary Fig. 2B close-up), suggested a median of >100 PNCTR molecules per PNC, significantly exceeding the RNA occupancy at *ACTB* transcription sites (Supplementary Fig. 2C, D).

Importantly, when we co-stained *ACTB* transcription sites by RNA-FISH with ACTB-Ex probes and HyPro-FISH (where HyPro-deposited biotinylation marks are visualized by fluorescent streptavidin[28]) with ACTB-Int probes, HyPro2 significantly outperformed the original enzyme (Fig. 1D–F). In particular, HyPro2 biotinylated a significantly larger fraction of RNA-FISH-positive transcription sites (Fig. 1E). Moreover, the intensity of biotinylated sites was significantly higher for HyPro2 compared to the original HyPro enzyme (Fig. 1F).

These results demonstrate that the modified HyPro enzyme significantly improves proximity labeling of a compartment containing just a few RNA molecules.

### Optimized conditions limit the diffusion of activated biotin while maintaining labeling efficiency

Enhanced proximity labeling of *ACTB* transcription sites highlighted another challenge in adapting the HyPro technology to small compartments: a detectable spread of the biotin label beyond the compartment boundaries delineated by RNA-FISH (arrows in Fig. 1D close-up). This spread likely resulted from the diffusion of activated biotin after its release from the peroxidase domain[40]. Given the small compartment size, such diffusion could substantially compromise labeling specificity by biotinylating a relatively large proportion of proteins not directly neighboring the RNA target.

The spread of biotin was also detectable when we proximity-biotinylated the PNC with the PNCTR-UC probe set and either the original HyPro or HyPro2, while simultaneously co-staining the compartment with the PNCTR-NR RNA-FISH probe (LVB samples in

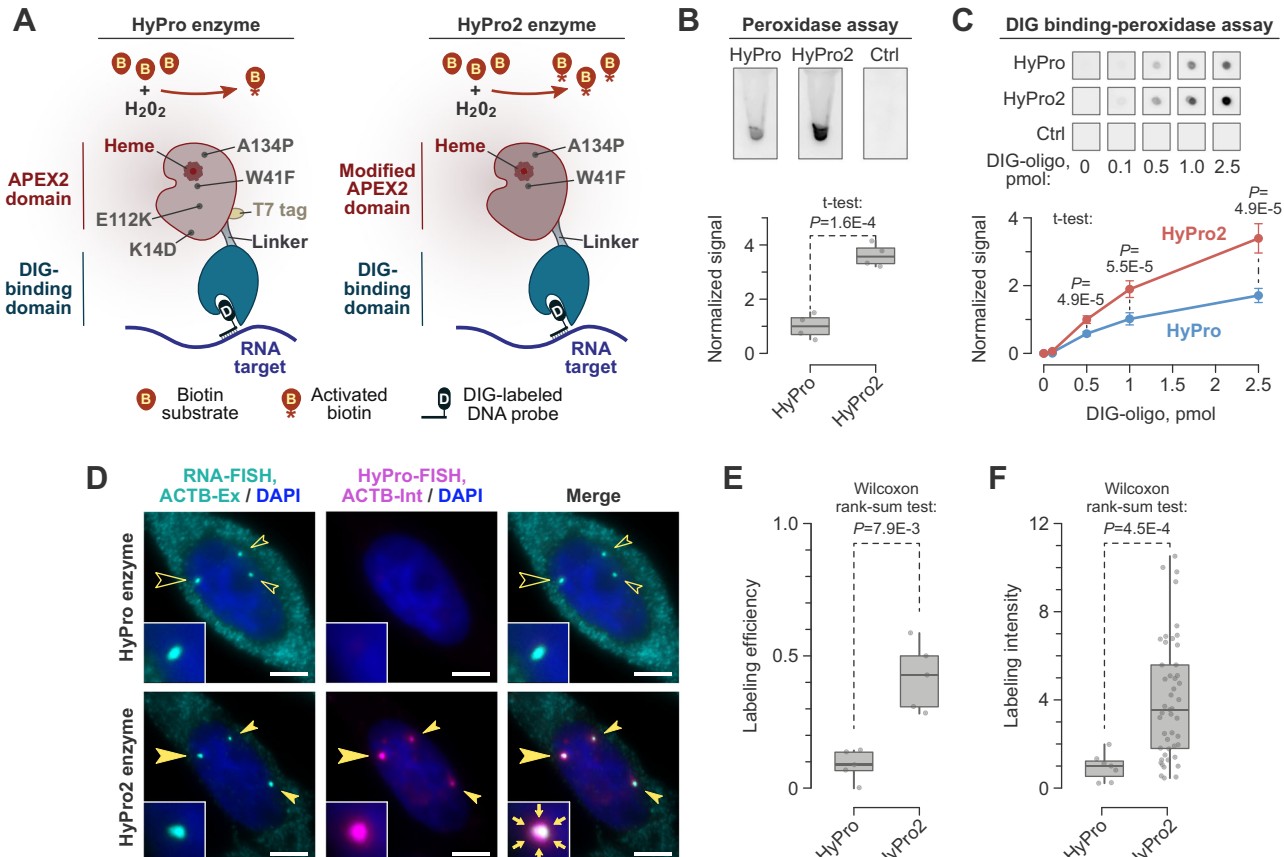

**Fig. 1 | HyPro2 enzyme enables efficient labeling of an RNA-containing micro-compartment. A** HyPro2 enzyme compared to its HyPro predecessor[28,42].
**B** Peroxidase activity of HyPro2 is higher than that of HyPro. Top, peroxidase chemiluminescence assays containing equal amounts of HyPro, HyPro2, or bovine serum albumin (Ctrl). Bottom, quantification of peroxidase activity from 4 experiments. Ctrl-subtracted data were normalized to the HyPro median, presented as box plots, and compared by a two-sided t-test, assuming unequal variance. **C** HyPro2 demonstrates superior activity in digoxigenin (DIG) binding-peroxidase spot assay, where HyPro or HyPro2 is recruited to a DIG-modified oligonucleotide immobilized on a nitrocellulose membrane (at 0 to 2.5 pmol per spot) and assayed for peroxidase-catalyzed chemiluminescence. Top, a typical result of this assay. Bottom, quantification of peroxidase activity from 6 experiments. The data were normalized to experiment-specific activity averages, and presented as mean ± SD, setting the 0-pmol values to 0. The statistical comparisons were made using a two-sided *t* test, assuming unequal variance. **D** HyPro2 but not HyPro allows efficient labeling of *ACTB* transcription sites. HeLa cells were co-stained with the

ACTB-specific exonic RNA-FISH probes (cyan) and the ACTB-specific intronic HyPro-FISH probes (magenta). HyPro-FISH labeling was performed for 1 min with either HyPro (top) or HyPro2 (bottom). RNA-FISH-positive *ACTB* transcription site foci are robustly biotinylated by HyPro2 (solid arrowheads) but not HyPro (open arrowheads). Foci marked by the large arrowheads are magnified 3 × in the close-ups. Arrows in the HyPro2 close-up indicate detectable diffusion of the biotin label outside of the RNA-FISH-positive area. Main images, maximum-intensity Z-stacks; close-ups, individual optical sections. Scale bars, 5 μm. **E** Proximity labeling efficiency, defined as the fraction of detectably biotinylated RNA-FISH-positive ACTB foci, is significantly higher for HyPro2 compared to HyPro. The box plot shows labeling efficiencies quantified from coverslip areas randomly selected from 3 labeling experiments. The data were compared by a two-sided Wilcoxon rank-sum test. **F** HyPro2-labeled foci are brighter compared to their HyPro-labeled counterparts. Individual *ACTB* transcription site labeling intensities were quantified from 3 experiments, presented as a box plot, and compared by a two-sided Wilcoxon rank-sum test.

---

Fig. 2A, B). To address the diffusion issue, we first increased labeling buffer viscosity by adding 50% sucrose, as suggested previously (TSA-seq[40]). While this approach successfully eliminated biotin diffusion, it caused a significant loss of HyPro and HyPro2 activity (sucrose samples in Fig. 2A, B and Fig. 2C, D).

Trehalose, another compound known to increase solution viscosity[41], proved to be a better alternative. Supplementing the labeling buffer with 50% trehalose, instead of sucrose, suppressed biotin diffusion with a significantly milder inhibitory effect on HyPro and HyPro2 activity (trehalose samples in Fig. 2A–D). Of note, HyPro2 gave rise to consistently stronger labeling compared to the original HyPro across all tested conditions, including those with trehalose-containing buffer (Fig. 2D).

While trehalose was less inhibitory than sucrose, we aimed to further enhance HyPro2 activity in the presence of trehalose to ensure efficient labeling of RNA microcompartments. Like other peroxidases, APX derivatives are heme-dependent enzymes. Although HyPro

purifies from *E. coli* in a heme-bound form[42], we wondered if supplementing the purified enzyme with additional heme might stimulate its activity. Indeed, pre-incubating HyPro and HyPro2 with the heme-related compound hemin significantly boosted their peroxidase activity in a concentration-dependent manner (Fig. 3A, B). Consistent with an earlier report[43], some background peroxidase activity was observed in negative control samples (Ctrl) pre-incubated with ≥ 20 μM hemin, but not at lower concentrations (Fig. 3A). Notably, HyPro2 retained its activity advantage over the original HyPro enzyme across the entire range of hemin concentrations (Fig. 3B).

These findings suggested the use of HyPro2 pre-incubated with < 20 μM hemin, along with the trehalose-containing buffer, for subsequent labeling experiments. A side-by-side comparison of proximity biotinylation patterns using HyPro2, with or without hemin pre-incubation (0, 5, and 15 μM), and either the low-viscosity buffer (LVB) or the trehalose-containing buffer, showed comparable labeling

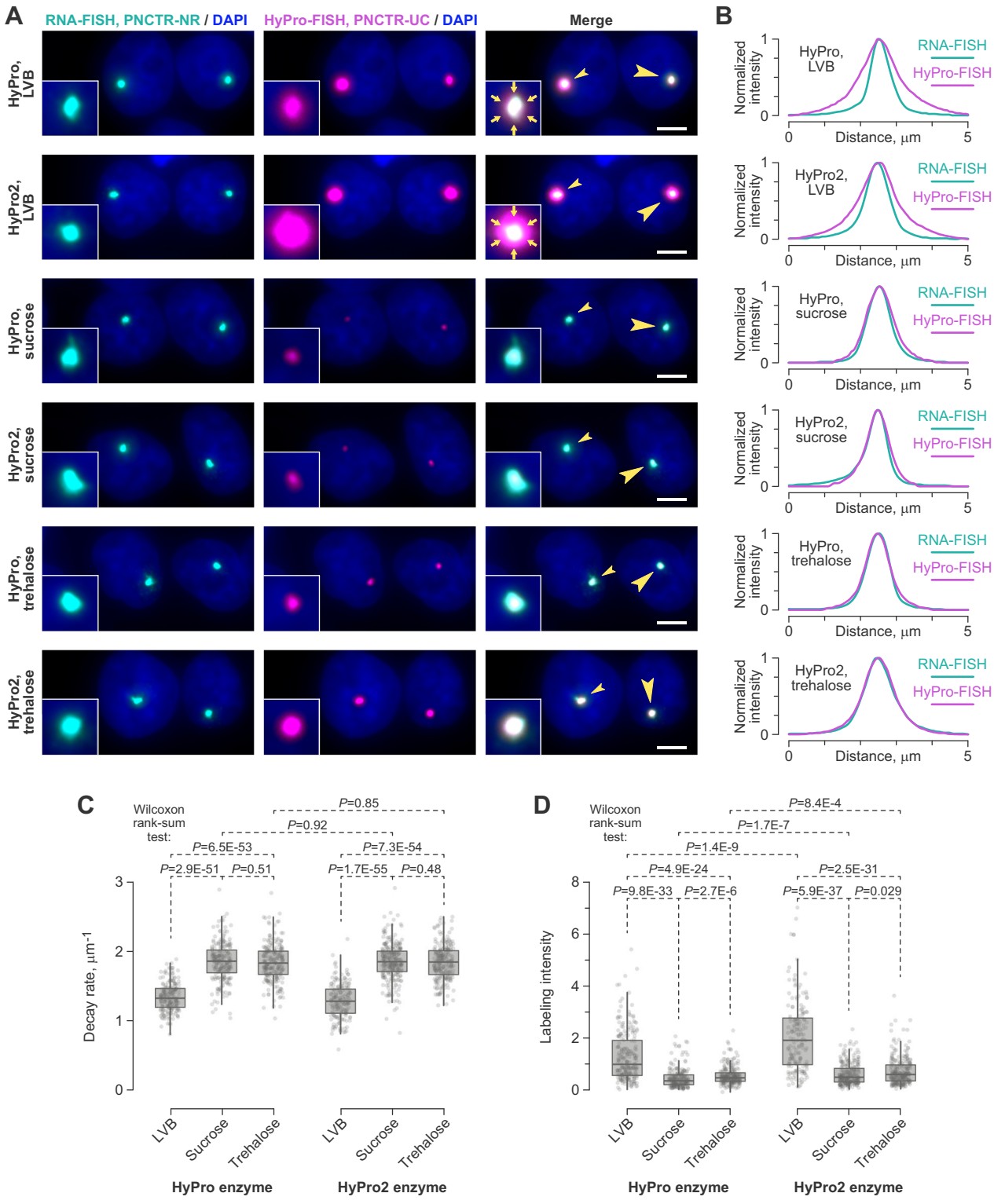

efficiency and intensity between the no-hemin/LVB and 5 μM hemin/trehalose conditions (Supplementary Fig. 3). The key advantage of 5 μM hemin/trehalose over no-hemin/LVB was the elimination of biotin spread beyond the RNA-FISH signal (Fig. 3C, D). Labeling efficiency and intensity further increased with 15-μM hemin/trehalose (Supplementary Fig. 3). However, this improvement was so pronounced that it became difficult to prevent over-saturation of *ACTB* transcription sites with biotin during the 1 min incubation period used throughout our study (Supplementary Fig. 3A). Interestingly, the nonspecific background staining occasionally observed in LVB was reduced under the

5 μM hemin/trehalose conditions (Fig. 3C), possibly due to the protein-stabilizing properties of trehalose[44].

This part of our work establishes optimized conditions for proximity biotinylation of RNA compartments, balancing efficient labeling with minimal signal diffusion.

## Enhanced labeling protocol successfully maps proteomes of compact RNA compartments

To test the ability of the optimized protocol to uncover compartment-specific proteomes, we labeled PNCs and *ACTB* transcription sites by

**Fig. 2 | Trehalose-containing buffer suppresses biotin diffusion while maintaining relatively high activity of proximity-labeling enzymes. A** The PNC compartment (arrowheads) was co-stained in HeLa cells by RNA-FISH (cyan) with a non-repetitive PNCTR-specific probe set and HyPro-FISH (magenta) with a probe targeting PNCTR's UC-rich repeats. Proximity biotinylation was performed in the low-viscosity buffer (LVB, rows 1 and 2) or a high-viscosity buffer containing either 50% sucrose (rows 3 and 4) or 50% trehalose (rows 5 and 6). Large arrowheads, perinucleolar compartments (PNCs) magnified 3 × in the close-ups. Main images, maximum-intensity Z-stacks; close-ups, individual optical sections. Scale bars, 5 μm. All samples were imaged using identical microscopy settings. Both HyPro (odd rows) and HyPro2 (even rows) perform well in this assay, with HyPro2 signals tending to be brighter. Sucrose and trehalose suppress biotin signal diffusion

(arrows in the LVB close-ups), with sucrose inhibiting enzymatic activity more than trehalose. The experiment was repeated 3 times, with similar results. **B** Maximum-normalized intensity profiles of RNA-FISH and HyPro-FISH signals for individual optical sections along the direction indicated by large arrowheads in (**A**). HyPro-FISH signal spreads beyond RNA-FISH in LVB but not in sucrose- or trehalose-containing buffers. **C** Signal decay rates of HyPro-FISH intensity profiles plotted as in (**B**) and fitted to an exponential decay model[40]. Larger decay rates in sucrose and trehalose indicate reduced signal diffusion. **D** Proximity-labeling efficiencies normalized to the HyPro-LVB median and compared across the six conditions shown in (**A**, **B**). Data in (**C**, **D**) are quantified from 3 labeling experiments, presented as box plots, and compared by a two-sided Wilcoxon rank-sum test.

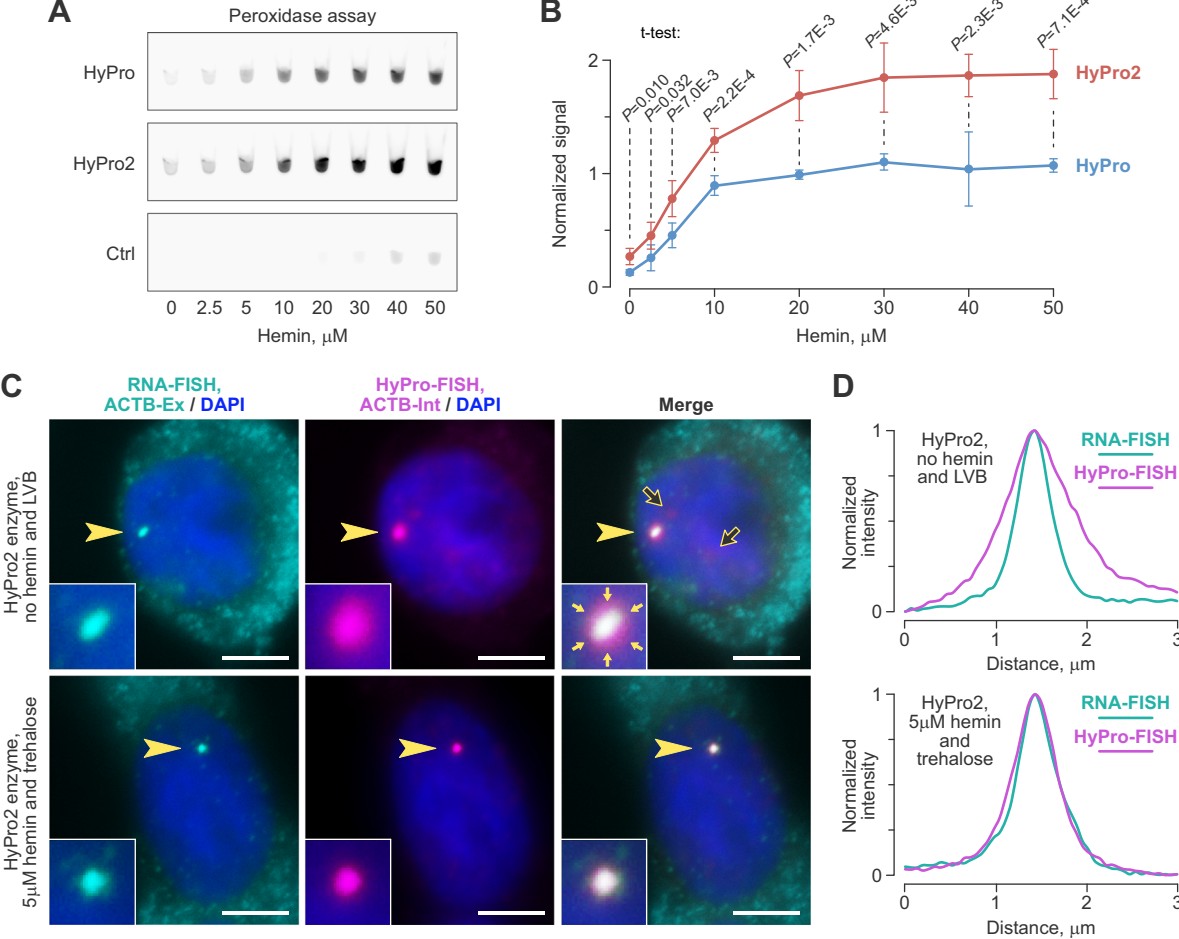

**Fig. 3 | Pre-incubation with hemin increases proximity-labeling efficiency. A** Hemin pre-incubation stimulates the peroxidase activity of the HyPro and HyPro2 enzymes. Note that negative control reactions (Ctrl) pre-incubated with 20–50 μM, but not lower concentrations of hemin, show background-level peroxidase activity[43]. **B** Quantification of the peroxidase activity data from (**A**) across 3 independent experiments. Data were Ctrl background-subtracted, normalized to experiment-specific averages, presented as mean ± SD and compared by a two-sided t-test assuming unequal variance. **C** Proximity labeling of *ACTB* transcription sites by HyPro2 in low-viscosity buffer (LVB) without hemin pre-incubation (top row) and in trehalose-containing buffer with 5 μM hemin pre-incubation (bottom row). *ACTB* transcription sites (arrowheads) are magnified 3 × in close-ups. Main

images and close-ups, individual optical sections. Scale bars, 5 μm. Both conditions were imaged using identical microscopy settings. Note that labeling intensities are comparable, but signal diffusion (filled arrows in the "no hemin and LVB" close-up) is effectively suppressed in the "5 μM hemin and trehalose" condition. Also note that hemin and trehalose suppress the nonspecific background staining occasionally detected in LVB (open arrows in the "no hemin and LVB" main image). The experiment was repeated 3 times, with similar results. **D** Maximum-normalized intensity profiles of RNA-FISH and HyPro-FISH signals along the direction indicated by large arrowheads in (**C**). HyPro-FISH signal spreads beyond RNA-FISH in "no hemin and LVB" but not in "5 μM hemin and trehalose".

hybridizing fixed and permeabilized HeLa cells with the PNCTR-UC or ACTB-Int probes, using the HyPro2 enzyme with 5 μM hemin/trehalose during the proximity-labeling step. Label-free mass spectrometry analysis of the biotinylated samples (enhanced HyPro-MS) showed that the PNCTR-UC and ACTB-Int probes labeled distinct sets of proteins compared to each other and to the no-probe control (Ctrl) (Fig. 4A).

The PNCTR-proximal proteome included known markers and cellular neighbors of the perinucleolar compartment (PNC), including RNA-binding proteins PTBP1, CELF1, and HNRNPL and DNA-associated MCM-complex subunits[28,29,45] (Fig. 4B and Supplementary Data 2). Consistent with the PNC proximity to the nucleolus[45], we also detected a subset of nucleolus-enriched proteins (e.g., GNL3, EBNA1BP2, NIFK,

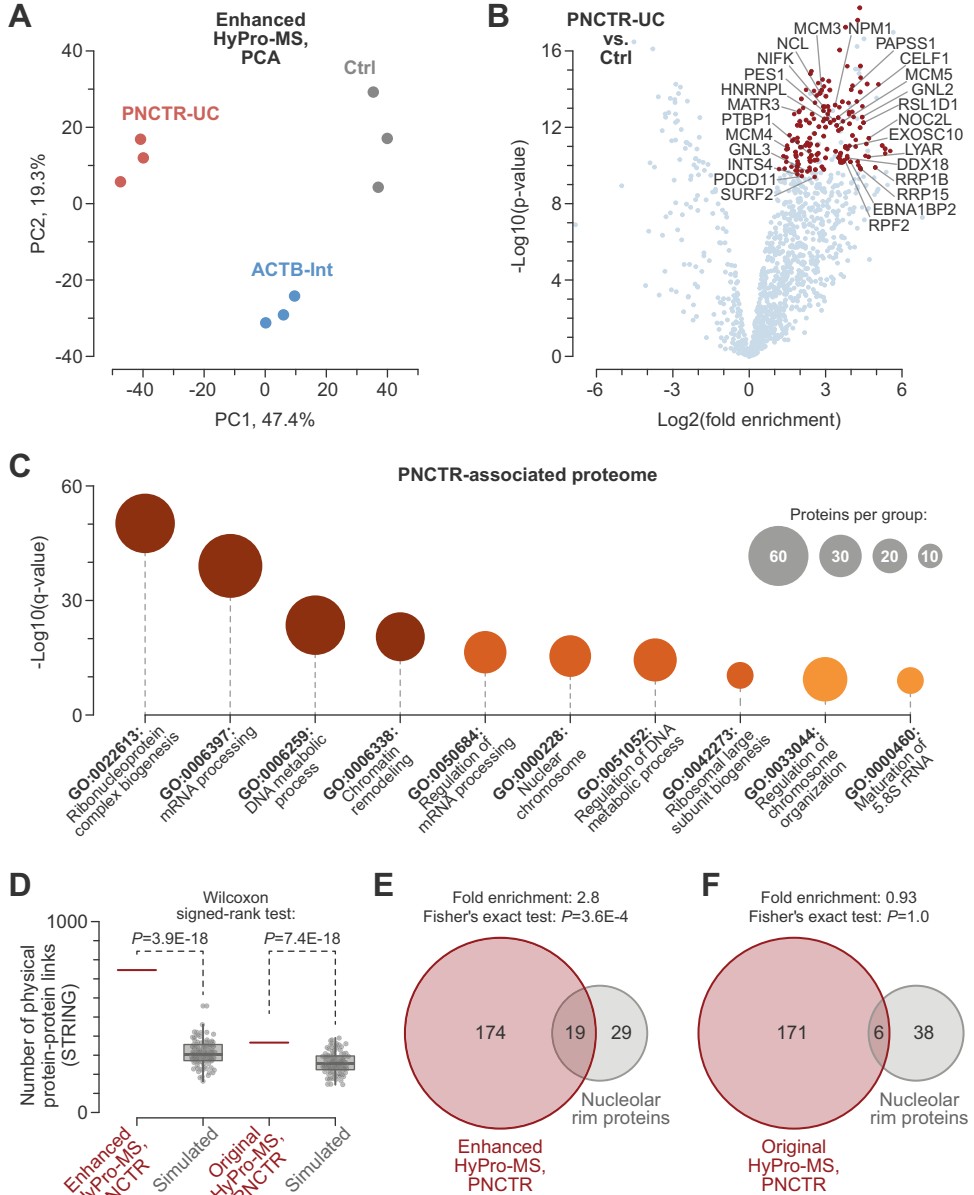

**Fig. 4 | Enhanced HyPro-MS enables specific labeling of the PNCTR-proximal proteome. A** Principal component analysis shows tight clustering of the PNCTR, ACTB and negative control (Ctrl) HyPro-MS samples. **B** Volcano plot for the PNCTR-proximal proteins known to localize to the nucleus based on the Human Protein Atlas data (https://www.proteinatlas.org/). Significant hits are shown in red. Known PNC markers and proteins localizing to the nucleolar rim compartment are labeled. **C** Metascape annotation of PNCTR-proximal proteome reveals enrichment for RNA and DNA metabolism functions, alongside aspects of ribosome biogenesis. **D** PNCTR-proximal proteomes identified by both the enhanced (this study) and the

original[28] versions of HyPro-MS are significantly enriched for physical protein-protein interactions from the STRING database[48] (score > 0.7), as compared to simulated (n = 100) controls, which are presented as box plots. However, the enrichment is markedly more robust for the enhanced HyPro-MS. The comparisons were made using a two-sided Wilcoxon signed rank test. **E** The PNCTR-proximal proteome identified by enhanced HyPro-MS is significantly enriched for nucleolar-rim proteins. **F** The PNCTR-proximal proteome mapped by the original HyPro-MS shows no significant enrichment for nucleolar-rim proteins. In (**E**, **F**), enrichment was assessed using a two-sided Fisher's exact test.

and PDCD11; Fig. 4B). Keeping this in mind, we focused on proteins known to localize to the nucleus in our subsequent analyses (see "Methods" for details). Functional annotation of PNCTR-proximal nuclear proteome in Metascape[46] revealed significant enrichment of proteins involved in both general RNA and DNA metabolism and specific steps of ribosome biogenesis in the nucleolus (Fig. 4C and Supplementary Data 3).

We evaluated the labeling specificity of the enhanced procedure by comparing the new data with the PNCTR-proximal proteome mapped by the original version of HyPro-MS[28] (Fig. 4D–F). Given that many RNA-associated proteins can interact with each other[47], we

reasoned that an increased HyPro-MS specificity should improve the detection of protein complexes. Indeed, STRING data[48] suggested that the 193 nuclear proteins identified by the enhanced PNCTR-UC HyPro-MS could form as many as 746 physical protein-protein interactions passing the recommended score > 0.7 cutoff (Fig. 4D). In comparison, only 366 such interactions were detected for the comparably sized nuclear proteome (177 proteins) identified by the original PNCTR-UC HyPro-MS using the same bioinformatics workflow (Fig. 4D).

Importantly, the number of interactions in both the enhanced and original shortlists was significantly greater compared to simulated proteomes, where the corresponding numbers of proteins were

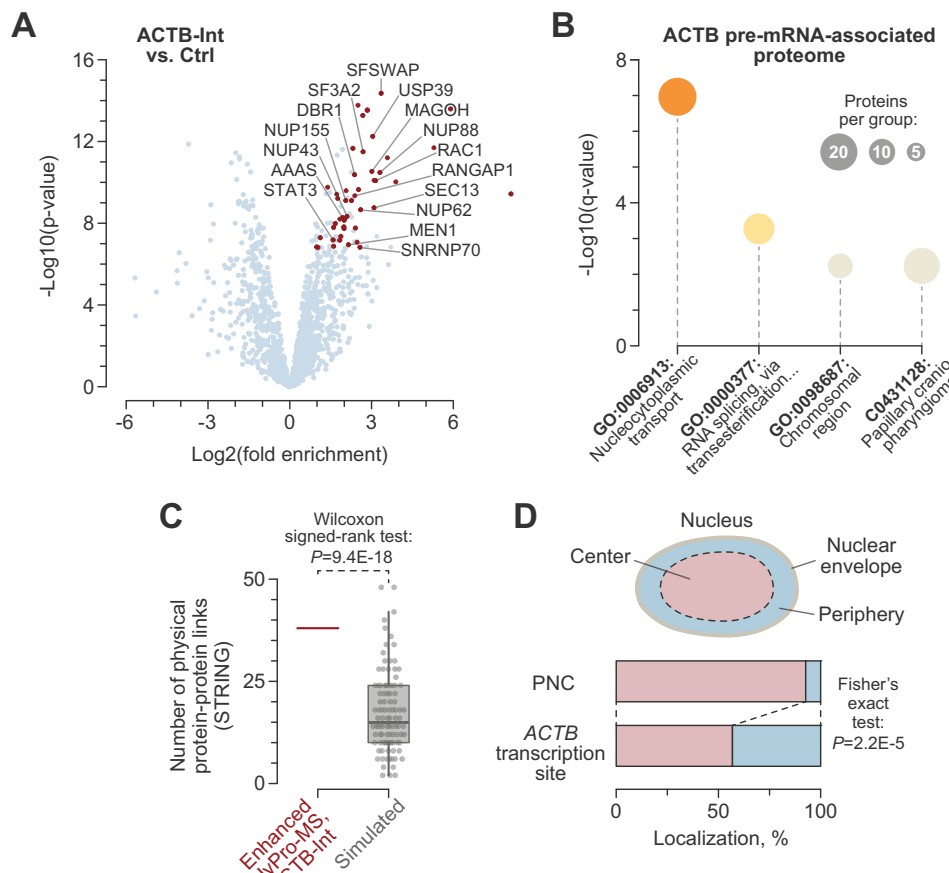

**Fig. 5 | Enhanced HyPro-MS identifies proteins associated with *ACTB* transcription sites. A** Volcano plots for the *ACTB* transcription site-proximal proteins known to localize to the nucleus based on the Human Protein Atlas (https://www.proteinatlas.org/) data. Significant hits are shown in red. Transcription, splicing and mRNA export factors, as well as nuclear pore components, are labeled. *P*-values are from a two-sided moderated *t* test[92], not adjusted for multiple testing. **B** Metascape analysis of the *ACTB* transcription site-proximal proteome demonstrates enrichment for nuclear transport and pre-mRNA export functions. **C** *ACTB* transcription site-proximal proteome is significantly enriched for physical protein-protein interactions from the STRING database[48] (score > 0.7), as compared to a simulated (*n* = 100; box plot) proteome control. The comparison was made using a two-sided Wilcoxon signed rank test. **D** Maximum-intensity projections of HeLa nuclei stained with either ACTB- or PNCTR-specific RNA-FISH probes were divided into central and peripheral regions of equal surface area. The distribution of *ACTB* transcription sites and PNCs between these two regions was compared using a two-sided Fisher's exact test. Note that *ACTB* transcription sites localize to the nuclear periphery more frequently than PNCs.

repeatedly sampled (*n* = 100) at random from the entire pool of mass-spec-detectable nuclear proteins. Yet, the enrichment of interacting protein pairs was noticeably more robust for the enhanced HyPro-MS (2.4-fold over the simulated median; *P* = 3.9E-18; two-sided Wilcoxon signed rank test) than for the original HyPro-MS (1.4-fold over the simulated median; *P* = 7.4E-18; two-sided Wilcoxon signed rank test) (Fig. 4D). Similar trends were observed when we used more relaxed (> 0.4) or more stringent (> 0.9) STRING score cutoffs (Supplementary Fig. 4A, B). Furthermore, the enhanced HyPro-MS fared strikingly better than its predecessor in identifying the interactions from the recently published RNA-aware protein-protein interactome (PPI) resource[47] (Supplementary Fig. 4C).

Of the several nucleolar sub-compartments, the outermost structure, known as the nucleolar rim[49], directly underlies the PNC located at the nucleolar periphery[45] (Supplementary Fig. 2D). Notably, the enhanced PNCTR-UC HyPro-MS proteome was significantly enriched for nucleolar rim-localized proteins (Fig. 4E). No such enrichment was detected amongst the original PNCTR-UC HyPro-MS data (Fig. 4F). Combined with the increased detection of protein-protein interactions (Fig. 4D and Supplementary Fig. 4), this points to a substantially improved ability of the enhanced HyPro-MS to identify immediate protein neighbors of a transcript of interest.

Inspection of the *ACTB* transcription site-proximal proteome (47 nuclear proteins; see "Methods" for details) revealed several factors involved in pre-mRNA splicing (e.g., DBR1, SF3A2, SNRNP70, SFSWAP, and USP39) and mRNA export from the nucleus to the cytoplasm (e.g., MAGOH, RANGAP1, SEC13, and several nucleoporins (NUPs)) (Fig. 5A and Supplementary Data 4). These observations were confirmed by Metascape (Fig. 5B and Supplementary Data 5). Interestingly, the disease-associated summary group "Papillary craniopharyngioma", which was also enriched in the Metascape analysis, comprised member terms associated with different aspects of tumor biology and the "Transcription by RNA polymerase II" (GO:0006366) function (Supplementary Data 5). A notable example of this functional category was the transcription factor STAT3, which is involved in the regulation of genes associated with cytoskeletal dynamics[50] and contains binding sites in the *ACTB*-overlapping enhancer region GH07J005523 (https://www.genecards.org[51];). Other examples included the STAT3 regulator RAC1[52] and the modulators of RNA polymerase II activity MEN1/Menin and RPAP1[53–55].

A surprising aspect of the *ACTB* transcription site-associated proteome was the abundance of nuclear pore-associated factors (Fig. 5A and Supplementary Data 5), including those known to regulate transcription (e.g., NUP62, NUP88, and SEC13) and those not

previously implicated in this process (e.g., NUP43 and NUP155)[56–59]. This contributed to a significantly higher than expected incidence of protein-protein interactions in the *ACTB* transcription site-associated proteome (Fig. 5C). To determine if the enrichment of nucleoporins might reflect the physical proximity of *ACTB* transcription sites to nuclear pores, we analyzed the distribution of these sites along the nuclear center-periphery axis in RNA-FISH-stained samples (see ACTB-Ex staining in Fig. 1D). This showed a significant bias of *ACTB* transcription sites toward the nuclear periphery, as compared to PNCTR/ PNC RNA-FISH (Fig. 5D; see PNCTR-NR staining in Fig. 2A).

Taken together, these data suggest that the enhanced HyPro protocol can identify proteomes of compartments containing a limited number of RNA molecules with high specificity and sensitivity.

## Enhanced HyPro technology is suitable for individual RNA molecules

To find out if the enhanced labeling procedure can be used to investigate even smaller RNA-containing compartments, we focused on endogenous C9orf72 transcripts with a C9-ALS/FTD-linked expansion of the G4C2 hexanucleotide repeat in the first intron (Supplementary Fig. 5A). We first characterized the expression of C9orf72 transcripts in induced pluripotent stem cells (iPSCs) derived from a healthy donor (C53) and a C9-ALS patient (DN19V4) containing ~638 copies of the G4C2 repeat in at least one *C9orf72* allele[60] (Supplementary Fig. 5A–C). In line with earlier studies[14,61,62], the total levels of C9orf72 mRNA detected by reverse transcription-quantitative PCR (RT-qPCR) with exon-2- and exon-3-specific primers (MLO3996/MLO3997; Supplementary Fig. 5A) were somewhat lower in DN19V4 than in C53 (Supplementary Fig. 5D). Since the expanded G4C2 repeat has been reported to promote the retention of the first intron in C9orf72 transcripts initiated at the upstream promoter[63] (V1 and V3 isoforms in Supplementary Fig. 5A), we included RT-qPCR analyses with primers amplifying the intron-1-exon-2 junction (MLO3994/MLO3995; Supplementary Fig. 5A). This revealed a ~6-fold increase in the abundance of intron 1-containing C9orf72 transcripts in DN19V4 compared to C53 (Supplementary Fig. 5E).

We then compared C53 and DN19V4 iPSCs using dual-color RNA-FISH with probes against C9orf72 exons (C9-Ex; a mixture of 48 antisense oligonucleotides) and G4C2 repeats (G4C2; an 18-mer with up to ~212 annealing sites in the expanded hexanucleotide repeat in DN19V4) (Fig. 6A and Supplementary Data 1). The main type of RNA signals detected in the C53 cells was C9-Ex-positive, G4C2-negative foci, which were evenly distributed across the nucleus and the cytoplasm (Supplementary Fig. 6A). Conversely, a large fraction of C9-Ex-positive foci in DN19V4 iPSCs was also G4C2-positive (Supplementary Fig. 6A). These double-positive foci predominantly localized to the nucleus, although some were occasionally observed in the cytoplasm. The C9-Ex-positive/G4C2-negative foci detected in DN19V4 iPSCs potentially originated from the wild-type *C9orf72* allele or represented productively spliced transcripts of the mutant *C9orf72* allele. In both C53 and DN19V4 cells, the G4C2 probe additionally detected a few C9-Ex-negative foci, which likely corresponded to nonspecific transcripts and were less abundant than the C9orf72-specific signals (Supplementary Fig. 6A).

Our FISH-quant analyses suggested that most C9-Ex-positive foci were single RNA molecules, regardless of the iPSC origin or the presence or absence of expanded G4C2 repeats in the transcripts (Supplementary Fig. 6B). Most of the remaining RNA-FISH foci contained two RNA molecules, with some of these signals potentially corresponding to *C9orf72* transcription sites (Supplementary Fig. 6B). Thus, the endogenous expression of *C9orf72* in iPSCs provided an excellent system for evaluating the sensitivity and biomedical utility of the enhanced HyPro technology (Supplementary Fig. 6C).

Combined staining of iPSCs with C9-Ex RNA-FISH and G4C2 HyPro-FISH using HyPro2 and the 5 µM hemin/trehalose conditions revealed robust biotinylation of a substantial fraction of C9-Ex-positive RNA signals in DN19V4 but not in C53 cells (Fig. 6B). Importantly, both single RNAs and small RNA clusters containing expanded G4C2 repeats were biotinylated with comparable efficiencies, with the single-RNA foci accounting for a majority of the total biotinylation signal (Fig. 6C, D).

Thus, the enhanced HyPro technique successfully biotinylates microcompartments containing single RNA molecules.

## Enhanced HyPro-MS illuminates early molecular events linked to C9-ALS pathology

To identify the protein interactome of nucleus-retained G4C2 repeat-containing transcripts originating from the mutant *C9orf72* allele, we analyzed DN19V4 iPSCs using enhanced HyPro-MS (using HyPro2 with 5 µM hemin/trehalose) with the G4C2 probe. The ACTB-Int probe set was used as a negative control to minimize contributions from nuclear proteins broadly associated with transcription sites. To control for nonspecific G4C2-labeling occasionally observed in healthy iPSCs (Fig. 6B), we additionally performed enhanced HyPro-MS with both G4C2 and ACTB-Int probes in the C53 iPSC line.

This approach identified 51 top-scoring candidates specifically associated with mutant C9orf72 transcripts in DN19V4 iPSC nuclei (Fig. 6E and Supplementary Data 6). These hits showed a robust overlap with proteins identified in a recent in vivo proximity-labeling study using a genetically engineered G4C2 repeat-containing "bait" ([20]; Fig. 6F). We additionally compiled a list of proteins consistently interacting with synthetic G4C2 repeat-containing transcripts in published studies[62,64–67] (Supplementary Data 7). Many of these proteins were also present in our G4C2 HyPro-MS data (Fig. 6G).

In line with the incompletely spliced nature of mutant C9orf72 transcripts (Supplementary Fig. 5E), Metascape analysis of the enhanced HyPro-MS hits revealed robust enrichment of RNA splicing-related functions (Fig. 6H and Supplementary Data 8). Another significantly enriched category included proteins linked to FTD and related neurodegenerative disorders, such as Pick's disease, frontotemporal lobar degeneration (FTLD), and ALS (Fig. 6H and Supplementary Data 8).

Many proteins in this disease-associated summary group have been studied as components of membraneless nuclear bodies known as paraspeckles[68]. "Paraspeckles" was also identified as an enriched Metascape summary group (Fig. 6H and Supplementary Data 8). Furthermore, proteins previously localized to paraspeckles using microscopy approaches[69,70] were significantly overrepresented in our shortlist (Fig. 6I). Importantly, several paraspeckle-associated proteins – including FUS, NONO, and SFPQ (Supplementary Data 6) – have been shown to associate with G4C2 repeat-containing transcripts in HEK293T overexpression models, as well as in fibroblasts and cerebellar cells from patients with C9orf72 mutations[71]. Furthermore, SFPQ has been directly implicated in the formation of G4C2 RNA foci[72].

To find out if mutant C9orf72 transcripts interact with paraspeckle proteins in pluripotent stem cells, we selected FUS and SFPQ – two paraspeckle markers with well-established links to ALS/FTD (Fig. 6E and Supplementary Data 8[69,71–76];) – for experimental validation. We co-stained DN19V4 iPSCs with RNA-FISH using G4C2 and C9-Ex probes, combined with immunofluorescence using FUS- and SFPQ-specific antibodies. Notably, C9orf72 transcripts with G4C2 hexanucleotide repeat expansion (HRE +) but not their normal counterparts (HRE-) frequently colocalized with or neighbored FUS- and SFPQ-positive protein densities dispersed across the nucleoplasm (Fig. 7A–D).

Importantly, a similar distribution was observed for SMARCC1/ BAF155 (Fig. 7E, F), another paraspeckle marker[70] (Supplementary Data 8), which – to the best of our knowledge – had not previously been linked to C9-ALS. No difference between the HRE + and HRE-transcripts was observed in their proximity to the RNA-binding protein

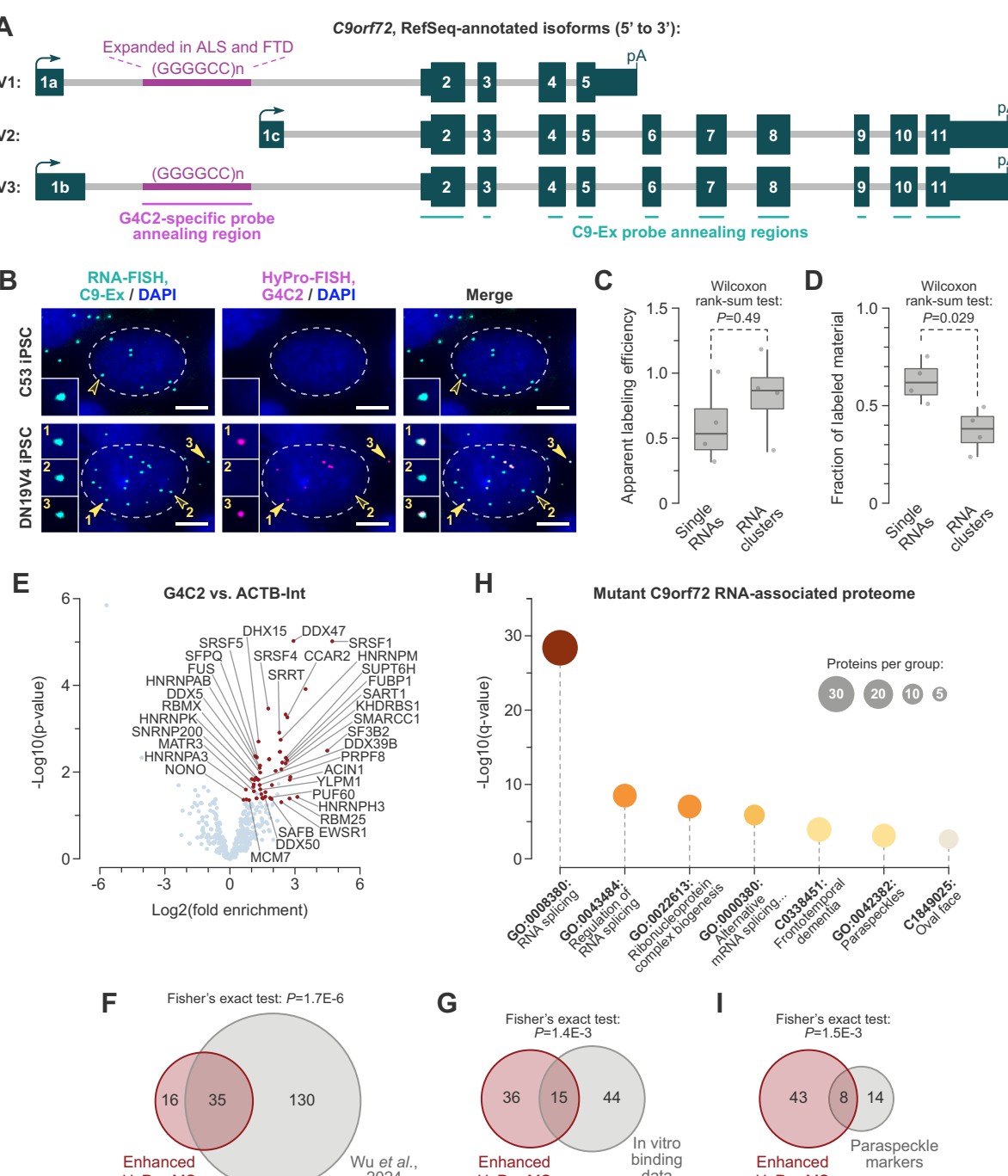

**Fig. 6 | Enhanced HyPro-MS reveals the interactome of endogenous C9orf72 transcripts with expanded G4C2 repeats. A** Diagram of the *C9orf72* gene and the probes used for RNA-FISH and HyPro-FISH experiments. **B** Induced pluripotent stem cells (iPSCs) derived from a healthy donor (C53) and a C9- ALS patient (DN19V4) were stained with C9orf72-specific exonic RNA-FISH probes (C9-Ex; cyan) and G4C2 repeat-specific HyPro-FISH probes (G4C2; magenta). Open arrowheads, G4C2 HyPro-FISH-negative C9orf72 transcripts, detectable in both C53 and DN19V4 samples. Solid arrowheads, G4C2 HyPro-FISH-positive C9orf72 transcripts, seen only in DN19V4 samples. Scale bars, 5 μm. **C** Apparent G4C2 HyPro labeling efficiency is statistically indistinguishable for RNA-FISH foci classified by FISH-quant as single RNAs or RNA clusters. Apparent efficiency was calculated by dividing the percentage of HyPro-labeled RNA-FISH foci in (**B**) by the percentage of the corresponding type of exonic probe-positive foci colocalizing with G4C2 staining in dual-color RNA-FISH experiments (Supplementary Fig. 6A). **D** Single RNAs account for a significantly larger fraction of G4C2 HyPro labeling compared to RNA clusters. Data

in (**C**, **D**) were quantified from coverslip areas randomly selected from 2 labeling experiments, presented as box plots, and compared by a two-sided Wilcoxon rank-sum test. **E** Volcano plot of proteins significantly enriched (red) near C9orf72 transcripts with expanded G4C2 repeats. Proteins identified in earlier studies using recombinant G4C2 repeat-containing transcripts are labeled. *P*-values are from a two-sided moderated *t* test[92], not adjusted for multiple testing. **F, G, I** Venn diagrams illustrating significant overlaps between the interactome of C9orf72 transcripts with expanded repeats and (**F**) proteins identified by genetically encoded proximity labeling of recombinant G4C2 repeat-containing transcripts[20]; (**G**) proteins consistently binding synthetic G4C2 RNA repeats in vitro (Supplementary Data 7); and (**I**) paraspeckles markers identified using microscopy approaches[69,70]. The data were analyzed using a two-sided Fisher's exact test. **H** Metascape annotation of the interactome of C9orf72 transcripts with expanded G4C2 repeats shows enrichment for RNA splicing factors, proteins implicated in neurodegenerative disorders (e.g., frontotemporal dementia, or FTD), and paraspeckle markers.

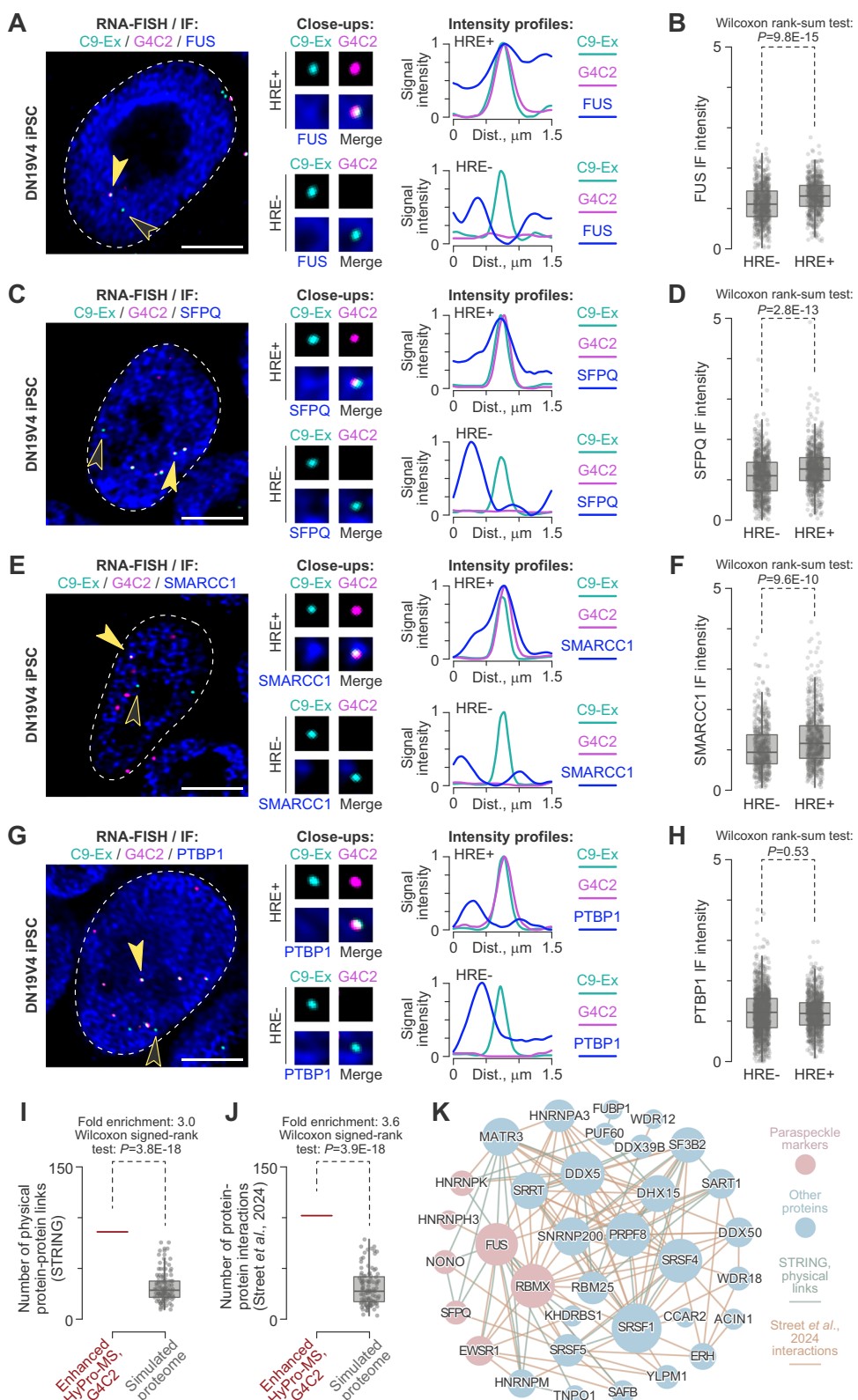

PTBP1, a negative control absent from the mutant C9orf72 proteome shortlist (Fig. 7G, H). Both previously uncharacterized (SMARCC1) and characterized (SFPQ) interactors formed preferential contacts with HRE+ transcripts in two additional C9-ALS iPSC lines, AST2 and M211R2 (Supplementary Figs. 7–9). This preference was not observed for the negative controls, PTBP1 and OCT4/ POU5F1, a well-known nuclear iPSC marker (Supplementary Figs. 8 and 9).

Given the limited information on the interaction between HRE + C9orf72 transcripts and SMARCC1, we further investigated this relationship using an orthogonal biochemical approach. To facilitate this, we constructed two *C9orf72* minigenes (*miniC9*): one lacking (*miniC9-CS*) and one containing (*miniC9-G4C2x100*) an expanded G4C2 repeat within the natural exon 1-intron 1-exon 2 context (Supplementary Fig. 10A). These constructs were expressed in wild-type C53 iPSCs, and

**Fig. 7 | Mutant C9orf72 transcripts colocalize with protein complexes containing paraspeckle markers in C9-ALS iPSCs. A,C,E,G** C9-ALS DN19V4 iPSCs were co-stained with RNA-FISH probes targeting C9orf72 exons (C9-Ex; cyan) and G4C2 repeats (G4C2; magenta), along with antibodies (blue) against paraspeckle markers: (**A**) FUS, (**C**) SFPQ, and (**E**) SMARCC1, or (**G**) PTBP1, an RNA-binding protein not associated with paraspeckles. Solid arrowheads, C9orf72 transcripts with G4C2 hexanucleotide repeat expansion (HRE + ); open arrowheads, C9orf72 transcripts without G4C2 hexanucleotide repeat expansion (HRE-). The marked transcripts are magnified 3 × in the close-ups. Signal intensity profiles on the right are plotted along 1.5 μm virtual lines drawn in the direction indicated by the arrowheads in the main images. For each cell, maximum channel-specific signal intensities are set to 1. HRE + but not HRE- transcripts colocalize with paraspeckle marker-positive densities. PTBP1-positive densities show no distinction between transcript types. Main images, deconvolved maximum-intensity Z-stacks; close-ups, deconvolved individual optical sections. Scale bars, 5 μm.
**B,D, F,H** Quantification of immunofluorescence signal intensity near HRE+ vs. HRE-

C9orf72 transcripts. Paraspeckle marker intensity is significantly higher near HRE + transcripts for (**B**) FUS, (**D**) SFPQ, and (**F**) SMARCC1. **H** No significant difference is observed for the PTBP1 control. Quantifications are from 3 staining experiments, presented as box plots, and compared by a two-sided Wilcoxon rank-sum test. **I, J** The G4C2 HyPro-MS proteome is enriched for physical protein-protein interactions from (**I**) the STRING database[48] (score > 0.7) and (**J**) the RNA-aware PPI[47] (including all types of interaction support). The comparisons were made using a two-sided Wilcoxon signed rank test. **K** A combined protein-protein interaction network built for the mutant (HRE + ) C9orf72 transcript-proximal proteome based on the STRING and RNA-aware PPI data suggests that the canonical paraspeckle markers[69] (red nodes) can form extensive physical contacts with each other and with pre-mRNA splicing factors and nuclear speckle components (most blue nodes). Node size is proportional to the number of links to other nodes. Links from STRING are colored cyan, and those from the RNA-aware PPI are colored orange. Mutant C9orf72 transcript-proximal proteins not connected to the network are omitted.

the resulting minigene-specific transcripts were analyzed by formaldehyde crosslinking and RNA immunoprecipitation using a SMARCC1-specific antibody (Supplementary Fig. 10B). An antibody against the known HRE + C9orf72 RNA interactor SFPQ and non-immune IgG served as a positive and negative controls, respectively (Supplementary Fig. 10B, C). RT-qPCR analysis of the immunoprecipitated fractions revealed that HRE+ miniC9-G4C2 x 100 transcripts interacted more efficiently with SMARCC1 and SFPQ compared to their HRE- miniC9-CS counterparts (Supplementary Fig. 10D–F). This effect was evident regardless of the normalization method used (Supplementary Fig. 10D, E). Omission of reverse transcriptase confirmed the specificity of the assay in detecting RNA-protein interactions (Supplementary Fig. 10F).

Finally, to systematically understand the relationship between the paraspeckle proteins and the numerous pre-mRNA splicing factors associated with mutant C9orf72 transcripts (Fig. 6E, H), we examined protein-protein interactions within the enhanced G4C2 HyPro-MS shortlist using both STRING[48] and RNA-aware[47] physical protein-protein interaction data (Fig. 7I–K). This analysis identified a strongly interconnected network, with the number of edges significantly exceeding those observed in randomly sampled proteomes (Fig. 7I, J). Remarkably, high-density connections were detectable not only within the splicing and paraspeckle protein modules but also between these two modules. This included, e.g., contacts between the paraspeckle proteins FUS, SFPQ, NONO, and RBMX and the SR protein SRSF5; between FUS and the splicing helicase DHX15; and between RBMX and the SR proteins SRSF1 and SRSF4, the spliceosomal components SNRNP200 and PRPF8, as well as the nuclear speckle marker RBM25 (Fig. 7K).

Overall, these results reveal the protein interactome of mutant C9orf72 transcripts endogenously expressed in patient-derived pluripotent stem cells, highlighting the utility of the enhanced HyPro-MS for single-molecule studies and advancing our understanding of early molecular events in C9-ALS.

## Discussion

The enhanced HyPro technology described in this study represents a significant advance in mapping protein interactomes of small, naturally occurring RNA-containing compartments and single RNA molecules. By modifying the previously designed HyPro enzyme to boost its peroxidase activity in vitro and optimizing labeling conditions with hemin and trehalose, we have lowered the detection limit of our method, enabling proteomics analyses of these challenging RNA targets. The improved performance of the HyPro2 enzyme is consistent with earlier observations suggesting that the K14D and E112K mutations can reduce APX activity under specific (though not all) experimental conditions, potentially by weakening heme incorporation as a result of decreased thermal stability of the enzyme[37].

The enhanced technology is similar to our original HyPro-labeling approach[28], in that it can be applied to genetically unperturbed samples with modest cell number requirements (0.5-1.0 × 10⁷ cells per enhanced HyPro-MS analysis). Compared to RNA-protein interactome mapping methods that rely on recombinant enzyme expression in living cells[77–79], both enhanced HyPro and its predecessor shorten experimental time, reduce toxicity and mislocalization artifacts associated with recombinant proteins, and broaden the range of biomedical applications.

What sets enhanced HyPro apart from earlier counterparts is its ability to rapidly generate a strong biotinylation signal with a limited number of target-specific probes, while efficiently limiting unspecific diffusion of activated biotin. For example, we used 34 intronic antisense oligonucleotides (ACTB-Int probe set; Supplementary Data 1), and only a 1 min incubation with the HyPro2 enzyme under 5 μM hemin/trehalose conditions, to specifically label *ACTB* transcription sites containing ~ 6 pre-mRNA molecules. Moreover, a 1 min incubation proved sufficient for robust detection of individual C9orf72 transcripts using a probe with a reasonable number (up to 212) of annealing sites in the expanded G4C2 repeat region. Increasing the hemin concentration at the pre-incubation step to 15 μM oversaturated the *ACTB* transcription site labeling signals (Supplementary Fig. 3). This suggests that our procedure can be readily adapted to investigate protein neighbors of relatively short RNAs or specific mRNA isoforms, where designing multiple probes per target would be impractical.

For comparison, the recently published O-MAP proteomics approach was tested for only highly abundant RNA targets: 47S, 7SK, and XIST[27]. Interestingly, microscopy imaging of lower-abundance RNAs (e.g., transgenic transcripts expressed from strong eukaryotic promoters or endogenous lncRNA Kcnq1ot1 and repeat-containing pre-mRNA WDR7) using O-MAP biotinylation required up to 120 min-long proximity-labeling reactions, using up to 200 target-specific probes and showing no evidence of single RNA molecule detection under these conditions[27]. This points to substantially greater sensitivity of enhanced HyPro-labeling compared to O-MAP. Moreover, the reliance of O-MAP on a low-viscosity labeling buffer (1 × PBS with 0.1% Tween-20), combined with extended reaction times, can result in unacceptably high background signals for microcompartments containing a limited number of RNA molecules.

We directly demonstrate that the enhanced HyPro technology can dissect protein interactomes of RNA compartments of varying sizes and complexities. Our proof-of-concept analyses identify both known and additional protein interactors, underscoring the potential of this approach for biomedical discovery. For instance, enhanced HyPro-MS successfully identified classical PNC markers PTBP1 and CELF1[45], as well as PNC-associated proteins, such as HNRNPL and MCM-complex subunits[28] (Fig. 4B). The newly mapped PNCTR interactome further included nucleolar rim proteins (Fig. 4B, E), which are expected to

neighbor the PNC in the cell (Supplementary Fig. 2D[45];). Notably, PNCTR analysis using the original HyPro method[28] did not detect CELF1 or the nucleolar rim signature (Fig. 4F), likely due to the "dilution" of immediate PNCTR neighbors by diffusive labeling of a more distant proteome in the low-viscosity buffer. Furthermore, proteomes identified by enhanced HyPro-MS were strongly enriched for protein-protein interactions (Figs. 4D, 7I, J, and Supplementary Fig. 4), a hallmark of RNA-associated protein networks[47].

The identification of splicing and transport factors in the *ACTB* transcription site-associated proteome (Fig. 5A, B and Supplementary Data 5) reinforces the notion of transcription site-proximal RNA processing in preparation for nucleocytoplasmic export[30–33]. Moreover, the association of *ACTB* transcription sites with nucleoporins suggests a degree of spatial coordination between RNA polymerase II transcription and nuclear export. This effect could result from (1) the physical proximity of the *ACTB* transcription site to a nuclear pore or (2) the recruitment of nuclear pore proteins to the *ACTB* transcription site as part of their non-canonical nucleoplasmic functions[56–59]. While future studies will be needed to distinguish between these possibilities, the preferential localization of *ACTB* transcription sites to the nuclear periphery (Fig. 5D) is consistent with the first scenario.

Our data suggest that, in C9-ALS patient-derived iPSCs, the mutant *C9orf72* allele with a genetically expanded G4C2 repeat gives rise to aberrantly spliced mRNA products. Evidence for bona fide retention of the G4C2-encoding intron 1 in these transcripts (Supplementary Fig. 5E), combined with their predominantly nuclear localization (Fig. 6B and Supplementary Fig. 6A), suggests that these transcripts are distinct from the alternative exon-1 isoforms recently identified in the cytoplasm of C9-ALS motor neurons[80]. Consistent with earlier studies[16], we demonstrate that the nuclear foci formed by G4C2 repeat-containing C9orf72 transcripts typically contain one RNA molecule (Supplementary Fig. 6).

Notably, most of the signal in our enhanced HyPro experiments corresponds to these single-molecule foci (Fig. 6D). By capturing RNA-binding proteins and nuclear neighbors specifically associated with these microcompartments in C9-ALS iPSCs, our results shed light on molecular mechanisms that may underlie early RNA processing and localization defects, occurring long before the onset of neuronal degeneration. In particular, we detect extensive contacts between mutant C9orf72 transcripts and nuclear proteins previously implicated in ALS/FTD pathology, including paraspeckle markers and pre-mRNA splicing factors (Fig. 6H, I and Supplementary Data 8). Notably, mutant C9orf72 transcripts have been shown to colocalize with a subset of paraspeckle-specific proteins in patient-derived fibroblasts and cerebellar cells[71,72]. A more recent study, also conducted in differentiated cells, identified an association between G4C2-containing transcripts and nuclear speckles[20], which neighbor paraspeckles in cells containing both types of these nuclear compartments[68].

Our work extends these observations by demonstrating that naturally expressed C9orf72 transcripts containing expanded G4C2 repeats come into contact with paraspeckle markers – including SMARCC1/BAF155, a SWI/SNF chromatin remodeling complex component[70] – already in pluripotent stem cells (Fig. 7A–H). Interestingly, these cells do not assemble bona fide paraspeckles and feature a distinctive nuclear speckle system[81–83]. The analysis of the protein-protein interaction network associated with mutant C9orf72 transcripts (Fig. 7I–K) suggests that, in this biological context, paraspeckle proteins can form complexes with pre-mRNA splicing factors and/or pluripotent stem cell-specific nuclear speckles. Future work investigating these possibilities could improve our understanding of molecular mechanisms at early, pre-symptomatic stages of ALS, potentially informing therapeutic strategies aimed at intervening before the clinical onset of the disease.

In conclusion, the enhanced HyPro technology enables sensitive, high-specificity profiling of protein interactomes of RNA-containing microcompartments and individual RNA molecules. We believe this tool has the potential to advance studies of diverse RNA-mediated processes in both basic and translational research.

## Methods

### Ethics statement
This study complies with international, national, and institutional ethical regulations. Work with human iPSC lines was approved by the UK National Research Ethics Service (REC reference 14/EM/1088). All iPSC lines used in this study were fully anonymized.

### Cell culture
HeLa cells (ATCC CCL-2) were maintained in a humidified incubator at 37 °C with 5% $CO_2$ in DMEM medium containing 4.5 g/L glucose, GlutaMAX™, and 110 mg/L sodium pyruvate (Thermo Fisher Scientific, cat# 11360070). The medium was supplemented with 10% fetal bovine serum (FBS; HyClone, cat# SV30160.03) and 100 units/ml Penicillin-Streptomycin (PenStrep; Thermo Fisher Scientific, cat# 15140122). For passaging, cells were washed with 1 × PBS and dissociated using 0.05% Trypsin-EDTA (Thermo Fisher Scientific, cat# 15400054) for 5 min at 37 °C.

Human induced pluripotent stem cells (iPSCs) derived from a healthy donor (C53)[84] or C9-ALS patients (DN19V4, AST2, and M211R2)[60] were cultured in a humidified incubator at 37 °C with 5% $CO_2$, in Essential 8 medium (Thermo Fisher Scientific, cat# A1517001) supplemented with 100 units/ml PenStrep, on plates coated with 1 μg/cm[2] Vitronectin-N (VTN-N; Thermo Fisher Scientific, cat# A14700). For passaging, cells were washed with DPBS (without calcium or magnesium; Thermo Fisher Scientific, cat# 14190094), incubated with 0.5 mM EDTA (Thermo Fisher Scientific, cat# 15575020) for 7 min at room temperature, and gently triturated in Essential 8 medium. Both HeLa cells and iPSCs displayed characteristic morphologies. iPSCs were further authenticated by assessing OCT4 expression and the G4C2 repeat expansion status in the *C9orf72* locus.

In *C9orf72* minigene transfection experiments, C53 cells were dissociated with Accutase (Thermo Fisher Scientific, cat# 00-4555-56) for 5 min at 37 °C, and the single-cell suspension (0.9-1.2 × 10^6 cells in total) was seeded into 10 cm dishes. On the following day, the cells were transfected with the appropriate constructs using Lipofectamine Stem Transfection Reagent (Thermo Fisher Scientific, cat# STEM00008).

### DNA constructs
DNA fragments used for cloning were amplified using either KAPA HiFi HotStart DNA Polymerase (Roche Diagnostics, cat# KK2502) or Platinum SuperFi II DNA Polymerase (Thermo Fisher Scientific, cat# 12361010). All restriction enzymes and T4 DNA Ligase (NEB, cat# M0202S) were from New England Biolabs. We used TOP10 *E. coli* (Thermo Fisher Scientific, cat# C404003) for routine cloning steps and either Stbl3 (Thermo Fisher Scientific, cat# C737303) or Stable (NEB, cat# C3040I) strains when dealing with repeat-containing constructs. Both pML752 (*miniC9-CS*) and pML753 (*miniC9-G4C2x100*) constructs, described below, were propagated in NEB Stable *E. coli* for consistency.

To facilitate stable maintenance of expanded repeats[85], we first inverted the origin of replication in the pEM157 cloning vector[86] using NEBuilder® HiFi DNA Assembly Master Mix (NEB, cat# E2621S) and the following primer pairs:

MLO4487, TCCACTGAGCGTCAGACCCCAGGAACCGTAAAAAGGCC and

MLO4488, CAAAAGGCCAGCAAAAGGCCGTAGAAAAGATCAAAGGATCTTC; and

MLO4489, GGCCTTTTGCTGGCCTTTTG and

MLO4490, GGGGTCTGACGCTCAGTG.

We then cloned a fragment of the *C9orf72* gene containing the alternative exon 1, intron 1 and constitutive exon 2 sequences into the

inverted-origin pML679 vector backbone in two consecutive steps: (1) The 5′ portion of the fragment was amplified from C53 iPSC genomic DNA using primers MLO4255 (GCTAGAATTCTGCGTCAAACAGCGA-CAA) and MLO4175 (GCTAACTAGTGACCTCCCTCCTGTTTCTGA), and inserted into pML679 at the EcoRI-SpeI sites. (2) The 3′ portion was amplified from the same genomic DNA using primers MLO4176 (GCTAACTAGTTGGGAATACTGCGGGTCTA) and MLO4177 (GCTA-CAATTGGAGTGTGGTTGGCAAGAAAAG), and inserted at the SpeI-MfeI sites. The wild-type (non-expanded) G4C2 sequences were then replaced with MluI-PstI cloning sites using the KLD site-directed mutagenesis with T4 DNA ligase, T4 Polynucleotide Kinase (NEB, cat# M0201S) and DpnI (NEB, cat# R0176S), resulting in pML752 (*miniC9-CS*).

The G4C2x100 sequence was generated using the RepEx-PCR method[85] with MLO4437 (/5Phos/GGGGCCGGGGCCGGGGCC) and MLO4438 (/5Phos/GGCCCCGGCCCCGGCCCC) primers. The repeat fragment was first cloned into the pGEM-T Easy vector (Promega), amplified using MLO4485 (GGTCTTCACGCGTGCGACGTCGGGCC-CAATTC) and MLO4486 (GAGACGCTGCAGTGGATGCA-TAGCTTGAGTATTCT) primers. The repeat-containing fragment was then subcloned into pML752 (*miniC9-CS*) at MluI-PstI, giving rise to pML753 (*miniC9-G4C4x100*). Sequences of both pML752 and pML753 were verified by whole plasmid sequencing (Full Circle Labs).

### Repeat-primed PCR genotyping
Genomic DNA was extracted from iPSCs using the GF-1 Tissue DNA Extraction Kit (Vivantis, GF-TD-050). Repeat-primed PCR was performed based on a previous study[14]. Each 15 µl reaction contained 1× SuperFi II Buffer, 1 unit of Platinum SuperFi II DNA Polymerase (Thermo Fisher Scientific, cat# 12361010), 0.2 mM each of 7-deaza-dGTP (New England Biolabs, cat# N0445S), dATP, dCTP, and dTTP (Roche, cat# 11969064001), 1 M betaine (Sigma-Aldrich, cat# B0300-1VL), 5% DMSO (Sigma-Aldrich, cat# D2650-100ML), 100 ng of genomic DNA, and 0.6 µM of each of the following primers:

MLO3654, CAGGAAACAGCTATGACC;
MLO3655, CAGGAAACAGCTATGACCGGGCCCGCCCCGACCACG CCCCGGCCCCGGCCCCGG;
MLO3683, /56-FAM/TGTAAAACGACGGCCAGTCAAGGAGGGAAA CAACCGCAGCC.

The PCR was carried out using a Bio-Rad C1000 Touch Thermal Cycler, with the ramp rate set to 0.5 °C/second. After an initial denaturation step at 98 °C for 10 min, a two-stage thermocycling program was used. In the first stage, the denaturation step was at 97 °C for 35 s, the primer annealing step was at 65 °C for 2 min, and the elongation step was at 68 °C for 8 min. This stage was run for 10 cycles. In the second stage, the denaturation and annealing steps were the same as in the first stage, but the elongation step was extended to 68 °C for 8 min and 20 s, with an additional 20 s added for each successive cycle. This stage was run for 25 cycles. A final elongation step was carried out at 68 °C for 7 min.

The PCR products were treated with ExoSAP-IT (Thermo Fisher Scientific, cat# 78250.40.UL) as recommended, followed by capillary-based fragment analysis by Genewiz.

### Reverse transcription-quantitative PCR (RT-qPCR)
Total RNA was extracted from iPSCs using the EZ-10 DNAaway RNA Mini-Preps Kit (Bio Basic, cat# BS88136) as recommended, and reverse-transcribed in 20 µl reactions. Prior to reverse transcription (RT), traces of genomic DNA were removed by treating 600 ng of total RNA with 1 µl of RQ1 RNase-Free DNase (Promega, cat# M6101) in a 9 µl reaction, which also contained RQ1 DNase buffer and 1 µl of murine RNase inhibitor (New England Biolabs, cat# M0314; 40 units/µl). The reaction was incubated at 37 °C for 45 min. RQ1 DNase was inactivated by adding 1 µl of the Stop Solution from the RQ1 kit and incubating the mixture at 65 °C for 10 min. The reaction was then placed on ice and

supplemented with either 1 µl of 100 µM random decamer (N10) or 1 µl of a mixture containing 4 µM each of the following C9orf72 and YWHAZ sense strand-specific RT primers:

MLO4000, TTATCAGGTCTTTTCTTGTT;
MLO4002, CACCAGTCGCTAGAGG;
MLO4035, GTAGCTGCTAATAAAGGTGA;
MLO4004, GGGAGTTCAGAATCTCATA;

followed by incubation at 70 °C for 10 minutes and subsequent cooling on ice.

Next, 1× buffer from the SuperScript IV kit, 10 mM DTT, 1 mM of each of the four dNTPs, 1 µl murine RNase inhibitor, and 1 µl of SuperScript IV Reverse Transcriptase (Thermo Fisher Scientific, cat# 18090200; 200 units/µl) were added to the final volume of 20 µl. In RT-minus controls, reverse transcriptase was replaced with nuclease-free water (Thermo Fisher Scientific, cat# AM9939). cDNA was synthesized by incubating the reaction at 50 °C for 40 minutes, followed by 70 °C for 10 min. The samples were then diluted 10-fold with 180 µl of nuclease-free water and stored at −80 °C until needed.

Quantitative PCR (qPCR) was performed in 20 µl reactions containing 10 µl of qPCRBIO SyGreen Master Mix Lo-ROX (PCR Biosystems, cat# PB20.11-51), 5 µl of diluted cDNA, and 100 nM of each primer. The following primer pairs were used:

*C9orf72* expression analysis of N10-primed cDNA:
MLO3996, CGCAGCACATATGGACTATCA;
MLO3997, TTCCATTCTCTCTGTGCCTTC;
*C9orf72* intron 1 retention analysis of cDNA primed by gene-specific RT primers:
MLO3994, CATTTGGGGTTTTGATGGAT;
MLO3995, GGTGATTTGCCACTTAAAGCA;
*YWHAZ* housekeeping gene expression analysis of cDNA primed by either N10 or gene-specific primers:
MLO3668, ACCGTTACTTGGCTGAGGTTGC;
MLO3669, CCCAGTCTGATAGGATGTGTTGG.

All reactions were performed on a LightCycler 96 Instrument (Roche). The thermocycling program included an initial DNA denaturation step at 95 °C for 120 s, followed by 45 cycles of 95 °C for 5 s and 60 °C for 20 s. Data were analyzed using LightCycler 96 software (Roche; version 1.1.0.1320).

### Formaldehyde-RNA immunoprecipitation assays
We used formaldehyde fixation–immunoprecipitation approaches described in earlier studies[87,88] with modifications. C53 iPSCs were transfected with minigene constructs in 10-cm dishes, cultured for 24 h, fixed with 1% formaldehyde (Thermo Fisher Scientific, cat# 28909) in 1× PBS for 10 min, and quenched with 125 mM glycine for 5 min at room temperature.

Cells were then washed twice with ice-cold 1× PBS, lysed in 300 µl of the lysis buffer (20 mM Tris-HCl, 150 mM NaCl, 10 mM EDTA, 1% NP-40, 0.1% SDS, 0.5% sodium deoxycholate, 1× protease inhibitor cocktail [Merck, cat# 04693132001], and 1 mM PMSF), and scraped into microfuge tubes. Lysates were sonicated using a Diagenode Bioruptor Plus for 10 cycles of 30 s on and 30 s off at high power. Sonicated lysates were centrifuged at 15,000 × g at 4 °C for 10 min, and the supernatants were transferred to new tubes.

Protein G Dynabeads (Thermo Fisher Scientific, cat# 10003D) were pre-washed twice with 1× PBS containing 0.1% Tween-20 and used for both lysate pre-clearing and immunoprecipitation steps. Lysates were pre-cleared by incubating them with the washed Protein G Dynabeads (0.3 mg per sample from each 10-cm dish) for 30 minutes at 4 °C. Beads were pelleted using a magnetic rack (Thermo Fisher Scientific, cat# 12321D), and the supernatants were transferred to new microfuge tubes.

For the immunoprecipitation step, 0.75 mg of washed Protein G Dynabeads were incubated with 6 µg of rabbit anti-SMARCC1 (Abcam, cat# ab172638; RRID: AB_3697426), rabbit anti-SFPQ (Abcam, cat#

38148; RRID:AB_945424), or non-immune rabbit IgG isotype control antibodies (Thermo Fisher Scientific, cat# 10500 C) for 1 h at 4 °C with rotation. The antibody-loaded beads were collected using the magnetic rack and washed twice with 1 × PBS containing 0.1% Tween-20.

Small aliquots (5%) of pre-cleared lysates were set aside as input controls. The remaining lysate was diluted with 4 volumes of 20 mM Tris-HCl, pH 7.5, 150 mM NaCl, and 5 mM EDTA and incubated with the antibody-loaded beads overnight at 4 °C with gentle rotation. Beads were then collected using the magnetic rack, washed twice with the original lysis buffer and twice with a high-salt buffer (20 mM Tris-HCl, pH 7.5, 500 mM NaCl, 5 mM EDTA, 0.5% sodium deoxycholate, 1% NP-40, 0.1% SDS, 1 mM DTT, 1 × protease inhibitor cocktail, and 1 mM PMSF), and resuspended in 106 µl of 20 mM Tris-HCl, pH 7.5.

Beads were then treated with 3 µl Turbo DNase (Thermo Fisher Scientific, cat# AM2238) and 1 µl recombinant RNase inhibitor (NEB, cat# M0314L) at 37 °C for 30 min. DNA digestion was stopped by adding 3.6 µl of 500 mM EDTA. Crosslinks were reversed by adding 4.8 µl of 5 M NaCl and 1.2 µl of 10% SDS, followed by incubation at 70 °C for 1 h. Subsequently, 6 µl of ~ 20 mg/ml Proteinase K (Thermo Fisher Scientific, cat# 10181030) was added, and the suspension was incubated at 55 °C for 30 min. Immunoprecipitated RNAs were extracted using Trizol-LS (Thermo Fisher Scientific, cat# 10296010) and purified with the Zymo RNA Clean & Concentrator kit (Cambridge Bioscience, cat# R1016), as recommended.

The purified RNAs were reverse-transcribed and analyzed by qPCR as described above, using C9mini-specific primers (MLO357: GCTACCGGACTCAGATCTCG and MLO4499: AGCAAGTAGTGGGGA-GAGAGG) and Neo-specific primers (MLO851: TGGATTCATC-GACTGTGGCC and MLO852: CGGTCATTTCGAACCCCAGA). MiniC9 RT-qPCR signals were normalized using the ΔCt method to either Neo marker mRNA levels (Supplementary Fig. 10D–E) or to minigene-specific Ct averages across all RT + pull-down samples, including non-immune IgG, anti-SMARCC1, and anti-SFPQ immunoprecipitations (Supplementary Fig. 10F). In Supplementary Fig. 10D, data were further normalized to the non-immune IgG control, while in Supplementary Fig. 10E, additional normalization was performed using input samples collected prior to immunoprecipitation.

### Expression and purification of HyPro and HyPro2 enzymes

The pML625 construct for HyPro2 expression in bacteria was derived from the HyPro expression plasmid pML433[28,42] (Addgene, #177190). To generate this construct, two amino acid positions in the APEX2 domain were reverted to the original soybean ascorbate peroxidase (APX) sequence (D14K and K112E) by stepwise QuikChange mutagenesis with KAPA HiFi DNA polymerase (Roche Diagnostics, cat# KK2101) and the following primers:

MLO3802, GTGGAATGGCACCTCTGGGCCACCTGTTAC;

MLO3803, GTAACAGGTGGCCCAGAGGTGCCATTCCAC;

MLO3804, CTTTTTGGCCTTTTCAACTGCTTTCTGATAGTCTGCG CTGACAG;

MLO3805, CTGTCAGCGCAGACTATCAGAAAGCAGTTGAAAAGG CCAAAAAG.

We additionally removed the N-terminal T7 expression tag present in the original HyPro enzyme using the following primers:

MLO2541, GTTTAACTTTAAGAAGGAGATATAGAATTCATGATTCC TCTGT;

MLO2542, ACAGAGGAATCATGAATTCTATATCTCCTTCTTAAAG TTAAAC.

SoluBL21 *E. coli* cells (Amsbio), transformed with either pML433 or pML625, were grown overnight in LB broth (VWR) containing 25 µg/ml kanamycin at 37 °C with shaking at 250 rpm. Four milliliters of the overnight culture were diluted into 600 ml of fresh LB broth containing 25 µg/ml kanamycin, and incubation was continued in a 2-liter flask at 37 °C with shaking until the culture reached an OD600 of 0.6 (~ 3 h). The cultures were then chilled on ice for 10 min, supplemented

with 1 mM isopropyl β-D-1-thiogalactopyranoside (IPTG, Promega), and shaken for 24 h at 25 °C to express the HyPro and HyPro2 proteins.

Cells were harvested by centrifugation at 10,000 × g for 10 minutes at 4 °C. The bacterial pellet was resuspended in 45 ml of BugBuster protein extraction reagent (Merck Millipore) supplemented with 1500 units/ml rLysozyme (Merck Millipore) and 25 units/ml benzonase (Merck Millipore) and incubated at room temperature for 30 min with constant rotation. The lysate was clarified by centrifugation at 16,000 × g for 20 min at 4 °C.

The supernatant was filtered through a 0.45 µm low-protein-binding syringe filter and loaded onto two sequentially connected 1 ml HisTrap FF Crude Columns (GE Healthcare) pre-equilibrated with buffer A (20 mM Tris, pH 8.0, 100 mM NaCl, 25 mM imidazole, and 14 mM β-mercaptoethanol (β-ME)). The column was washed with 20 ml of buffer A, and proteins were step-eluted with 50% buffer B (20 mM Tris, pH 8.0, 100 mM NaCl, 500 mM imidazole, and 14 mM β-ME).

The eluted fractions were then loaded onto a HiLoad 26/60 Superdex 75 column (GE Healthcare) equilibrated with buffer C (20 mM Tris, pH 8.0, 100 mM NaCl, and 1 mM DTT). Protein elution was monitored by UV absorbance at 280 nm, and protein concentrations in the fractions were measured using a Pierce BCA Kit (Thermo Fisher Scientific, cat# 23225) as recommended. Fractions containing the highest concentrations of HyPro and HyPro2 and showing no major contaminating protein bands on Coomassie R-250-stained SDS-PAGE gels were pooled, aliquoted, snap-frozen in liquid nitrogen, and stored at −80 °C for up to one year.

### Peroxidase activity assays

To assess the peroxidase activity of HyPro and HyPro2 enzymes, 1 µl of 2.7 µg/µl protein sample was mixed with 10 µl of reconstituted Enhanced Chemiluminescence (ECL) substrate (Thermo Fisher Scientific, cat# 32109) in thin-wall 0.2 ml PCR tubes (Axygen, cat# AX-PCR-0208-CP-C-CS). The mixture was incubated for 1 min and subsequently imaged using an Odyssey FC system (LI-COR). Bovine serum albumin (BSA) served as a negative control. In some experiments, proteins were pre-incubated with 0–50 µM hemin-HCl (Sigma-Aldrich, cat# H9039-1G) for 10 minutes at room temperature before assaying their activity.

To evaluate both digoxigenin binding and peroxidase activities of HyPro and HyPro2, serial dilutions of digoxigenin-labeled PNCTR-UC oligonucleotides (Supplementary Data 1; and see below) were spotted onto a 0.45 µm nitrocellulose membrane (Sigma-Aldrich, cat# GE10600016) and UV-crosslinked at 120 mJ/cm² using a Stratalinker. The membrane was rinsed with 1 × TBST buffer (20 mM Tris-HCl, pH 7.6, 150 mM NaCl, and 0.2% Tween 20), blocked with 1 × TBST containing 5% BSA at room temperature for 1 hour, and then incubated with HyPro or HyPro2 in 1 × TBST containing 1% BSA at room temperature for 1 hour. The membrane was finally washed three times with 1 × TBST, soaked in reconstituted Immobilon ECL reagent (Millipore, cat# WBKLS0500), and imaged using an Odyssey FC system (LI-COR).

### DNA probes for RNA-FISH and hybridization-proximity labeling

DNA oligonucleotide probe sets (Supplementary Data 1) were designed using the Stellaris® probe designer program (LGC Biosearch Technologies, v4.2). For RNA-FISH experiments, we typically used Quasar 570- or Quasar 670-labeled probe sets from Biosearch Technologies. The G4C2-specific probe was ordered from IDT with either the /3AlexF488N/ or the /3AlexF647N/ modification.

For hybridization-proximity labeling and some RNA-FISH experiments, unmodified oligonucleotides purchased from IDT were labeled using a 2nd-generation DIG Oligonucleotide 3′-End Labeling Kit (Sigma-Aldrich; cat# 3353575910) to yield 5 µM digoxigenin-labeled mixtures. Some oligonucleotides, including the repeat-specific PNCTR-UC and G4C2 probes (Supplementary Data 1), were ordered from IDT with a 3′-terminal digoxigenin modification (/3Dig_N/).

## RNA-FISH and immunofluorescence

Cells grown on 18 mm coverslips (VWR International, cat# HECH1015/18) were rinsed once with 1 × PBS and fixed with 4% formaldehyde (ThermoFisher Scientific; cat# 28908) for 15 min at room temperature. The coverslips were then washed three times with 1 × PBS for 5 min each, permeabilized with 70% ethanol for 1 h at room temperature or overnight at 4 °C, and rinsed again with 1 × PBS.

For experiments combining RNA-FISH with immunofluorescence, fixed and permeabilized cells were blocked with IF blocking buffer (1 × PBS supplemented with 1% BSA, 0.2% Tween-20, and 20 units/ml murine RNase inhibitor) for 30 min at room temperature. The cells were then incubated with an appropriate primary antibody diluted in IF blocking buffer for 1 hour at room temperature or overnight at 4 °C. We used the following primary antibodies: rabbit anti-FUS (Proteintech, cat# 11570-1-AP; RRID:AB_2247082; 1:200 dilution), rabbit anti-OCT4 (Abcam, cat# ab19857; RRID: AB_445175; 1:500 dilution), rabbit anti-PTBP1 (Abcam, cat# ab133734; RRID:AB_2814646; 1:250 dilution), rabbit anti-SFPQ (Abcam, cat# ab38148; RRID:AB_945424; 1:300 dilution), and rabbit anti-SMARCC1 (Abcam, cat# ab22355; RRID:AB_2191988; 1:500 dilution).

Following incubation with primary antibodies, the coverslips were washed three times with 1 × PBS and incubated with Alexa Fluor-conjugated secondary antibodies (Alexa Fluor488-conjugated anti-rabbit IgG (H + L) (ThermoFisher Scientific, cat# A-21206; RRID: AB_2535792; 1:300 dilution) or Alexa Fluor647-conjugated anti-mouse IgG (H + L) (ThermoFisher Scientific, cat# A31571; RRID:AB_162542; 1:300 dilution)) for 1 h at room temperature. This was followed by three additional washes with 1 × PBS, signal fixation with 4% formaldehyde for 15 min at room temperature, and three final washes with 1 × PBS.

RNA-FISH staining was performed after completing these optional immunofluorescence steps. Coverslips were typically washed with 2 × SSC containing 10% formamide and incubated overnight at 37 °C with 100–400 nM antisense oligonucleotide probes diluted in hybridization buffer (2 × SSC, 10% formamide, and 10% dextran sulfate). After hybridization, the samples were washed at 37 °C with 2 × SSC containing 10% formamide for 30 min, followed by a wash in 1 × SSC at room temperature for 15 min. Somewhat more stringent hybridization and washing conditions were used for the G4C2 repeat-specific probe, as explained in the Hybridization-proximity labeling section.

For fluorophore-labeled probes, the samples were subsequently rinsed with 1 × PBS, stained with 0.5 µg/ml DAPI in 1 × PBS for 3 min at room temperature, and mounted onto microscope slides using ProLong Gold Antifade Reagent (ThermoFisher Scientific; cat#P36934). When digoxigenin probes were used for RNA-FISH, the 1 × SSC washing step was followed by blocking the coverslips in 4 × SSC supplemented with 1% BSA and 100 units/ml murine RNase inhibitor at room temperature for 30 min. The coverslips were then incubated with mouse anti-digoxigenin antibody (Jackson Laboratories, cat# 200-002-156; RRID:AB_2339005; 1:500 dilution) in the same buffer overnight at 4 °C. After incubation, the coverslips were washed sequentially with 4 × SSC, 4 × SSC containing 0.1% Triton-X100, and 4 × SSC for 10 minutes each. Subsequently, they were incubated with an Alexa Fluor-conjugated anti-mouse secondary antibody in IF blocking buffer for 1 h at room temperature or overnight at 4 °C. For biotin-labeled RNA-FISH probes, used in some of our optimization experiments, the signal was detected using Alexa Fluor647-conjugated streptavidin (Biolegend, cat# 405237; 1:500 to 1:200 dilution) in IF blocking buffer for 1 h at room temperature.

The samples were then washed sequentially with 4 × SSC, 4 × SSC containing 0.1% Triton-X100, and 4 × SSC for 10 miutes each. They were briefly rinsed with 1 × PBS, stained with 0.5 µg/ml DAPI in 1 × PBS for 3 min at room temperature, and mounted onto microscope slides using ProLong Gold Antifade Reagent (ThermoFisher Scientific; cat#P36934). Images were acquired using a ZEISS Axio Observer Z1

inverted microscope equipped with an alpha Plan-Apochromat 100 ×/ 1.46 oil immersion objective. Z-stacks were captured at 0.22 µm intervals. Routine image analysis was done using ZEISS ZEN 2.5 Pro (https://www.zeiss.com/microscopy/en/products/software/zeiss-zen. html), FIJI[89], and Cellpose 2.2.3[90]. Images in Fig. 7 and Supplementary Figs. 8 and 9 were deconvolved using "Constrained Iterative" algorithm of ZEISS ZEN 2.5 Pro, with the point spread function experimentally determined using 0.1 µm TetraSpeck Fluorescent Microspheres (Thermo Fisher Scientific, cat# T7284).

To estimate the number of RNA molecules per compartment, we used FISH-quant (v2)[39]. Briefly, RNA-FISH spots were first detected using the detection.spots_thresholding() function with manually adjusted intensity threshold to include single molecules. We then used the detection.decompose_dense() function to convert RNA-FISH spots into single-molecule equivalents and the detection.detect_clusters() function to allocate single molecules to individual compartments. The identities of single molecules in dual-color RNA-FISH experiments were determined using the multistack.detect_spots_colocalization() function.

## Hybridization-proximity labeling

Cells grown in 12-well plates with 18-mm coverslips (1-2 × 10⁵ cells per well) or 10-cm dishes (0.5-1.0 × 10⁷ cells per dish) were fixed with 0.5 mg/ml dithiobis(succinimidyl propionate) (DSP; Thermo Fisher Scientific, cat# 22585) in 1 × PBS for 30 min at room temperature. The cells were then washed three times with 1 × PBS and 20 mM Tris-HCl, pH 8.0, for 5 min per wash and permeabilized with 70% ethanol at room temperature for 1 h. We typically equilibrated fixed and permeabilized samples in 2 × SSC and 10% formamide (Thermo Fisher Scientific) for 5 min and hybridized them overnight at 37 °C with 50–200 nM (adjusted based on RNA target abundance) of digoxigenin-labeled probes in 2 × SSC, 10% formamide, and 10% dextran sulfate. For the G4C2 repeat-specific probe, hybridizations were carried out in 2 × SSC, 15% formamide, and 10% dextran sulfate at 42 °C.

After hybridization, the samples were washed with 2 × SSC containing 10% formamide at 37 °C (or 15% formamide at 42 °C for the G4C2 repeat-specific probe) for 30 minutes, followed by 1 × SSC at room temperature for 15 min. Samples were then blocked with 0.8% BSA in 4 × SSC (HyPro blocking buffer) supplemented with 100 units/ml murine RNase inhibitor at room temperature for 30 min.

Next, the samples were incubated with 2.7 µg/ml HyPro or HyPro2 enzyme, with or without hemin pre-incubation, in HyPro blocking buffer at room temperature for 1 h. Unbound enzyme was removed by sequential washes with 4 × SSC, 4 × SSC containing 0.1% Triton-X100, and 4 × SSC for 10 min each, followed by a single rinse with 1 × PBS. The samples were then equilibrated in 1 × PBS, 50% sucrose in 1 × PBS, or 50% trehalose in 1 × PBS for 5 min.

Proximity biotinylation was performed by adding an equal volume of 1 × PBS, 50% sucrose, or 50% trehalose containing 1 mM biotin-phenol (Caltag Medsystems, cat# CDX-B0270) and 0.2 mM hydrogen peroxide (Sigma-Aldrich, cat# H1009), followed by gentle agitation for 1 min at room temperature. The reaction was quenched by rapidly rinsing the samples three times with 5 mM Trolox (Sigma-Aldrich, cat#238813) and 10 mM sodium ascorbate (Sigma-Aldrich, cat#A4034) in 1 × PBS. Samples on coverslips were used for HyPro-FISH, while those processed in dishes were analyzed by mass spectrometry.

## HyPro-FISH

Proximity-labeled samples on 18-mm coverslips, prepared as described above, were rinsed with 1 × PBS and 4 × SSC, then incubated with A647- (Biolegend, cat# 405237; 1:500 dilution) or A555-conjugated streptavidin (Thermo Fisher Scientific, cat# S32355; RRID:AB_2571525; 1:500 to 1:200 dilution) in HyPro blocking buffer supplemented with 80 units/ml murine RNase inhibitor at room temperature for 1 h. The samples were washed

sequentially with 4 × SSC, 4 × SSC containing 0.1% Triton-X100, and 4 × SSC for 10 minutes per wash.

After washing, the samples were briefly rinsed in 1 × PBS, stained with 0.5 μg/ml 4′,6-diamidino-2-phenylindole (DAPI) in 1 × PBS for 3 min at room temperature, and mounted onto microscope slides using ProLong Gold Antifade Reagent (Thermo Fisher Scientific, cat# P36934). In experiments where HyPro-FISH was combined with RNA-FISH, HyPro-FISH-labeled samples were typically post-fixed with 4% PFA for 10 min, washed with 1 × PBS and stained using the RNA-FISH procedure described above.

Images were acquired using a ZEISS Axio Observer Z1 inverted microscope equipped with an alpha Plan-Apochromat 100 ×/1.46 oil immersion objective. Z-stacks were captured at 0.22 μm intervals.

## Purification of biotinylated proteins

Proximity-labeled cells in 10-cm dishes were lysed in 600 μl of high-SDS RIPA buffer supplemented with 10 mM sodium ascorbate, 5 mM Trolox, 50 mM DTT, cOmplete EDTA-free protease inhibitor cocktail (Sigma-Aldrich, cat# 4693132001), and 1 mM phenylmethanesulfonyl fluoride (PMSF; Cell Signaling Technology, cat# 8553). Lysates were incubated on ice for 10 min, scraped from the plates, and incubated for an additional 10 min on ice. Samples were then sonicated using a Diagenode Bioruptor Plus for 10 cycles (30 s on, 30 s off) at high power and de-crosslinked by incubation at 37 °C for 1 h. The lysates were clarified by centrifugation at 15,000 × g for 10 min at 4 °C, transferred to fresh tubes, and stored at −80 °C until further use.

Sixty microliters of MyOne streptavidin C1 magnetic beads were pre-washed twice with RIPA buffer (150 mM NaCl, 1 mM EDTA, pH 8.0, 50 mM Tris-HCl, pH 8.0, 1% NP-40, 0.5% sodium deoxycholate, and 0.1% SDS), resuspended in 3 ml of RIPA buffer, and combined with de-crosslinked lysates in three separate aliquots. The mixture was incubated at room temperature for 2 h with gentle agitation.

After incubation, the beads were collected using a DynaMag™−2 Magnet and washed sequentially to remove nonspecifically bound proteins: twice with RIPA buffer, once with 1 M KCl, once with 0.1 M Na$_2$CO$_3$, once with 2 M urea in 10 mM Tris-HCl, pH 8.0, and twice more with RIPA buffer. Finally, the beads were collected using the DynaMag™−2 Magnet and analyzed by mass spectrometry as described below.

## Label-free mass spectrometry

Protein-loaded beads were washed three times with 50 mM ammonium bicarbonate (pH 8.0) and resuspended in 45 μl of 50 mM ammonium bicarbonate (pH 8.0) containing 1.5 μg of Trypsin/Lys-C protease mix (Promega). On-bead proteolysis was performed by incubating the suspension overnight at 37 °C with agitation. The following day, an additional 0.75 μg of Trypsin/Lys-C in 15 μl of 50 mM ammonium bicarbonate (pH 8.0) was added to the suspension, and the incubation was continued for another 4 h at 37 °C.

After digestion, the beads were collected using a DynaMag™−2 Magnet, and the hydrolysates were transferred to fresh microfuge tubes. The beads were washed twice with 45 μl aliquots of mass spectrometry-grade water, and these washes were combined with the original supernatants, bringing the final volume to ~150 μl and the ammonium bicarbonate concentration to ~20 mM. The samples were cleared by centrifugation at 16,000 × g for 10 min at 4 °C and transferred to fresh tubes. All conditions were analyzed in triplicate, processing labeled samples in parallel (technical triplicates).

The subsequent sample preparation and label-free mass spectrometry steps were performed by the CEMS Proteomics Core Facility at King's College London, UK. Peptides were purified using Pierce C18 spin columns (Thermo Fisher Scientific, UK) according to the manufacturer's instructions, eluted in 70% acetonitrile, and dried in a SpeedVac (Thermo Fisher Scientific, UK). The samples were then resuspended in 2% acetonitrile in 0.05% formic acid (both from Fisher Scientific, UK) and analyzed by LC-MS/MS.

Chromatographic separation was performed using a U3000 UHPLC NanoLC system (Thermo Fisher Scientific, UK). Peptides were resolved by reversed-phase chromatography on a 50 cm-long, 75 μm I.D. C18 PepMap column using a linear gradient formed by buffers A (0.1% formic acid) and B (80% acetonitrile in 0.1% formic acid). The gradient was delivered at a flow rate of 250 nL/min, starting at 5% buffer B (0–5 min), gradually increasing the percent of buffer B to 40% (5–40 min), followed by a 99% buffer B wash (40–45 min), and re-equilibrating the column at 5% buffer B (45–60 min).

The eluates were ionized by electrospray ionization using an Orbitrap Fusion Lumos Tribrid mass spectrometer (Thermo Fisher Scientific, UK) controlled by Xcalibur v4.1.5 software. The instrument was programmed to acquire data in data-dependent mode using a "universal" Orbitrap-Ion Trap method, with a 3-second cycle time between a full MS scan and MS/MS fragmentation. Orbitrap spectra (FTMS1) were collected at a resolution of 120,000 over a scan range of m/z 375-1500, with an automatic gain control (AGC) setting of 4 × 10$^5$ and a maximum injection time of 35 ms. Monoisotopic precursor ions were filtered by charge state (+2 to +7) with an intensity threshold set between 5.0 × 10$^3$ and 1 × 10$^{20}$, and a dynamic exclusion window of 35 s with ±10 ppm. MS2 precursor ions were isolated in the quadrupole with a mass width filter of 1.2 m/z. Ion trap fragmentation spectra (ITMS2) were collected with an AGC target setting of 1 × 10$^4$, a maximum injection time of 35 ms, and CID collision energy set at 35%. This method takes advantage of the multiple analyzers on the Orbitrap Fusion Lumos, utilizing all available parallelizable time and minimizing dependence on method parameters.

## Mass spectrometry data analyses

Raw mass-spec data files were processed using Proteome Discoverer (v2.2; Thermo Fisher Scientific, UK) to search against Uniprot Swis-sprot *Homo sapiens* Taxonomy (49,974 entries) using Mascot (v2.6.0; www.matrixscience.com) and the Sequest search algorithms[91]. Precursor mass tolerance was set to 20 ppm with fragment mass tolerance set to 0.8 Da with a maximum of two missed cleavages. Variable modifications included carbamidomethylation (Cys) and oxidation (Met). Searching stringency was set to 1% False Discovery Rate (FDR). In total, 3056 proteins were detected in HeLa samples and 1439 proteins were detected in iPSC samples.

Label-free quantification (LFQ) intensity data were imported into R. Keratins and proteins appearing in > 50% of all the control experiments in the CRAPome database (https://reprint-apms.org/), with > 15 average spectral count among the protein-positive control samples, were excluded from further analyses. The filtered data were then analyzed using the DEP package (v1.26.0)[92]. We also filtered the data to include proteins only identified in all 3 replicates of at least one condition (either compartment-specific or control HyPro-MS) and used default imputation settings (fun = "MinProb", q = 0.01).

Proteins enriched in the comparison between a specific probe and control ≥1.5-fold with a P-value < 0.05 and a P-value-based Z-score >1 were shortlisted for further analysis. When comparing the PNCTR-UC and ACTB-Int HyPro-MS data, we additionally required that compartment-specific Z-scores exceeded the Z-scores of the same proteins in the other compartment. Similarly, when shortlisting proteins neighboring C9orf72 transcripts with expanded G4C2 repeats, we required that Z-scores for C9-ALS iPSCs (DN19V4) exceeded the Z-scores for the healthy control (C53).

Given that a subset of incompletely spliced ACTB transcripts and expanded G4C2 repeat-containing C9orf72 transcripts can escape to the cytoplasm (e.g., see Fig. 6B), we excluded HyPro-MS hits potentially interacting with target transcripts in the cytoplasmic compartment from downstream bioinformatics analyses. To achieve this, known nuclear proteins were shortlisted using subcellular localization

data from the Human Protein Atlas (https://www.proteinatlas.org/), applying filters for nuclear localization and nuclear components (e.g., nucleoli, nuclear bodies, nucleoplasm, etc.) as either the "main" or "additional" localization. Functional annotation of target RNA-associated proteins was performed using Metascape[46] (v3.5; https://metascape.org), incorporating GO Biological Processes, GO Cellular Components, and DisGeNET terms, while maintaining QC as part of the enrichment calculations.

## Statistics and reproducibility

All statistical analyses were performed in R. Biochemical and gene expression data were analyzed using a two-sided Student's t-test, assuming unequal variance. Quantified microscopy images were compared using a two-sided Wilcoxon rank-sum test. The numbers of edges in protein-protein interaction networks were compared with those in simulated proteomes using a two-sided Wilcoxon signed-rank test. Fisher's exact test was applied for protein set enrichment analyses. The number of experimental replicates, the statistical tests used, and the corresponding P-values are provided in the Figures and/or Figure legends. Data reproducibility was confirmed by performing at least 2, and typically ≥ 3, independent experiments. No statistical method was used to predetermine sample size, and no data were excluded from the analyses. The experiments were not randomized, and the investigators were not blinded to allocation during experiments and outcome assessment. Unless indicated otherwise, data are shown as mean with error bars representing ± SD. In all box plots, the box bounds indicate the first and third quartiles, and the horizontal line within the box represents the median. Whiskers extend from the quartiles to the minimum and maximum data points, or to 1.5 times the interquartile range in the presence of outliers.

## Reporting summary

Further information on research design is available in the Nature Portfolio Reporting Summary linked to this article.

## Data availability

The proteomics data generated in this study have been deposited to PRIDE as projects PXD063191 and PXD063192. Uncropped microscopy images have been uploaded to figshare (https://doi.org/10.6084/m9.figshare.29282240)[93]. Other types of. Source data are provided in this paper.

## Code availability

Publicly available computer programs used in this study are described in the Methods section. In-house image analysis code is available through figshare (https://doi.org/10.6084/m9.figshare.29282240)[93].

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

## Acknowledgements

We thank Siddharthan Chandran, Ivo Lieberam, Snezhka Oliferenko, Tatyana Shelkovnikova, and Selina Wray for reagents and valuable discussions, and Steven Lynham for assistance with mass spectrometry analyses. This work was supported by the Biotechnology and Biological Sciences Research Council (grants BB/Y009304/1, BB/V006258/1, and BB/R001049/1 to E.V.M.), the Medical Research Council (MR/Z506187/1 to E.V.M.), and the Motor Neurone Disease Association (Shaw/Mar19/893-792 and Nishimura/Dec20/942-793).

## Author contributions

Conceptualization: K.Y., T.H.C. and E.V.M. Methodology: K.Y., T.H.C., E.C.H., A.L.N., C.E.S. and E.V.M. Investigation: K.Y., T.H.C. and E.V.M. Visualization: K.Y., T.H.C., and E.V.M. Funding acquisition: A.L.N., C.E.S., and E.V.M. Project administration: E.V.M. Writing - original draft: K.Y., T.H.C. and E.V.M. Writing - review and editing: K.Y., T.H.C., E.C.H., A.L.N., C.E.S. and E.V.M. Supervision: C.E.S. and E.V.M.

## Competing interests

The authors declare no competing interests.
