## [Transparent Peer Review file · Nature Communications]

Enhanced hybridization-proximity labeling discovers protein interactomes of single RNA molecules

Corresponding Author: Professor Eugene Makeyev

Version 0:

Reviewer comments:

Reviewer #1

(Remarks to the Author)

Controlling the function of mRNAs requires their assembly with specific sets of proteins. Identifying these proteins and their RNA binding dynamics, both locally and temporally, is of great interest to those seeking to understand cellular RNA function. Several proximity labeling techniques have been developed in recent years to characterize the interactome ('proxisome') of specific RNAs. Limitations have been the sensitivity (i.e., the bait RNA had to be of minimal abundance) of the approach, the diffusion of the labeling biotin residue (leading to labeling of unwanted bystanders), or the need for in-cell expression of recombinant versions of proximity labeling enzymes, e.g., as part of fusion proteins that target them to the RNA of interest. The original HyPro technique developed by the Makeyev lab (Mol Cell) has allowed to overcome the second hurdle, but still suffers from the problem of low sensitivity. In this manuscript, the group describes an improved version of HyPro (eHyPro) that overcomes the sensitivity and spreading problems while allowing the use of proximity labeling methods on RNA-associated proteins in unmodified cells.

Major findings:

By carefully studying (and improving) the original HyPro enzyme and labeling procedure, the authors develop an optimized protocol for HyPro-based proximity labeling. This is first demonstrated with in vitro and in cell bulk biotinylation, and then demonstrated on beta-actin RNA transcription sites (TS) and perinucleolar compartments (which are enriched in PNCTR RNA) by detecting biotinylation using fluorescence microscopy. Importantly, the actual proximity labeling experiments for both beta-actin TS and perinucleolar compartments to detect RNA-associated proteins by mass spectrometry also suggest that eHyPro outperforms the original HyPro method. More importantly, they demonstrate for a nuclear (pre-splicing) RNA that eHyPro allows proximity labeling of RNAs with only a few molecules present.

Major point of criticism:

Many of the proteins identified as residing in (what the authors call) RNA-proximal compartments or RNA neighborhoods have already been associated with the RNA under study. This makes other proteins identified in these RNA-proximal compartments very likely candidates for modulators of the RNA in question. However, beyond cross-correlation with existing data on these proteins and their interaction in association networks, no additional experimental support for the association of these new factors (such as the paraspeckle markers identified as novel C9orf72 transcript associates) is provided. I suggest to approach this by an orthogonal experimental setup like superresolution microscopy or co-IP.

Minor points:

In line 106, the authors claim that the engineered versions of APEX (APEX, APEX2) have reduced activity compared to the original APX enzyme. These should be checked again. Comparing the $k(\text{cat})/K_m$ values of APX, engineered APEX, and APEX2 (Martell, Nat. Biotech. 2012; Lam et al., Nat. Meth. 2015), the engineered versions show an increase but no decrease. APX (W41F) by 8.2-fold, mAPX (K14D, E112K) by 1.5-fold, APEX (i.e. mAPX+W41F) by 8.1-fold. This also means that the authors observe an increase in activity from HyPro to eHyPro ok BUT in the original description (Martell, Nat. Biotech) there is no reduction of K14D / E112K compared to wt APX. Perhaps the authors can explain these contrasting observations in the Discussion?

Figure 1D: The eHyPro Biotin image shows additional signals that are not detected by FISH in the corresponding partner images. Does this mean that eHyPro can label RNAs that are not detectable by smFISH or that eHyPro can also lead to

false positive signals? Regarding the experimental setup with hemine and trehalose (e.g., in Figure 3C), why do these conditions, which are expected to limit diffusion around 'positive' spots, also seem to reduce the number of such putative false-positive signals?

Figure S2B and line 138: The image quality of the image showing a diffraction-limited spot used to quantify the number of individual PNCTR molecules does not appear to be sufficient to verify whether it really reflects a spot, as it appears too blurred.

Figure S3A and S3B: What is the difference between labeling efficiency and labeling intensity? The main text and figure legend do not clearly state what these are and why both are shown.

Figure S6A: The figure shows 3 cells (or 3 nuclei) with a very uneven distribution of signals/dots between them. Is this imbalance a general phenomenon (i.e., is there a large variation in foci between cells of the same type)?

Figure 6B and lines 310/311: The authors state that "...conditions revealed robust biotinylation of a substantial fraction of C9-Ex-positive RNA signals..." The labeling of the Y-axis suggests values of 0.5 - 1%. Can this really be called 'substantial'?

Figure 7 A to G: It is not clear from the main text and figure legends how exactly the intensity plots were derived. For example, the figure legend states that 'The signal intensity profiles on the right are plotted along the directions indicated by the arrowheads in the main figures'. If the profile is plotted along a virtual line between the arrowheads, this does not seem to fit the intensity distribution shown.

Reviewer #2

(Remarks to the Author)

This is an interesting methods paper that presents a new approach to detect protein interactomes of selected RNAs. The paper is well written and in step by step modification of their protocol is an improvement on the available techniques. However, it falsely lays claim to novelty regarding paraspeckle component in G4C2 repeat interactome, ignoring/avoiding/diminishing already published findings. This is a major issue and needs to be addressed/corrected.

L35. 'Uncovered' needs to be changed to 'confirmed'. Using 'uncovered' is misleading as the technique only confirms an already published paraspeckle-like properties of the G4C2 repeat (Bajc Cesnik et al, J Cell Biol 2019) and further confirmed by Malnar et al., J Cell Biol 2021.

L37. 'reveal' is also an overclaim, because the findings are known.

L104. Consider renaming eHyPro as it is odd to name something as enhanced, while the process is actually reverting the mutations introduced in APEX2 and removing a T7 tag.

L335. 'gratifyingly' is not a scientific term. In the following lines it should be mentioned that Bajc Cesnik paper needs mentioning

L343-351. These experiments are just a repeat of what has already been shown in Bajc Cesnik for patient fibroblasts and in case of SFPQ, patient brains. This should be mentioned.

L448. The only mention of the Bajc Česnik paper is here and it is very vague. It should be mentioned that the study differentiated cells are C9orf72 patient fibroblasts. It is completely omitted that there is data on colocalization in C9orf72 brains. Also the manuscript should mention that SFPQ regulates G4C2 foci formation (Malnar et al. J Cell Sci 2021)

L453-456. The statement is misleading. What is the proof that the clusters of paraspeckle proteins are small? Also, G4C2 foci are NEAT1 independent (See Bajc Cesnik, J Cell Biol 2019), therefore there is no reason why there should be some mystery in the interaction of G4C2 with paraspeckle proteins in the situation where there are no paraspeckles.

Reviewer #3

(Remarks to the Author)

Multiple proximity labeling strategies have been described that are designed to identify the proteins bound to specific RNAs. Here, Yap et al report a modification of their hybridization-based proximity (HyPro) labeling protocol, or enhanced (e)HyPro, to optimize the identification of proteins interacting with single RNA molecules. Modification of the HyPro APEX2 domain to increase peroxidase activity and DIG binding resulted in eHyPro labeling of ACTB transcription sites using both exon and intron probes. Since biotin label diffusion was observed with eHyPro, they next limit this diffusion using 50% trehalose while also demonstrating hemin preincubation (5 M) enhances proximity labeling. PC analysis of eHyPro MS results showed distinct clustering of control, PNCTR and ACTB proteome profiles with multiple RBPs and nucleoli-associated proteins identified in the PNCTR proteome, including integrator (INTS4) and exosome (EXOSC10) proteins, together with a similar analysis of ACTB transcription sites that also highlighted several NUP proteins which led the authors finding a nuclear periphery bias of these sites. Finally, they identify the C9orf72 RNA interactome in control vs C9-ALS iPSCs and identify interactions between single RNA foci, paraspeckle and splicing proteins. Overall, this is a careful and well-written study that

reports a technical advancement in the proximity labeling field, but I have a few concerns noted below.

1. Abstract, ln 28. The authors describe eHyPro for proteomic mapping of RNA microcompartments (<10 RNA molecules) but designate their prime example, C9orf72 mutant iPSCs with expanded G4C2 repeats, as 'genetically unperturbed cells'. This is confusing when referring to a genetic mutant and it might be clearer to simply state that the iPSCs are only expressing endogenous RNA transcripts.
2. Results, ln 333. Fig. S6E does not exist – I assume the authors were referring to S6C.
3. Fig. 6. This figure describes a proteomic composition analysis based on only single male healthy and c9-ALS iPSCs. First, these results should be confirmed using iPSCs from multiple donors since these are readily available. Second, ALS affects primarily motor and cortical neurons, and thus a more disease relevant analysis would be to identify the protein composition associated with C9 RNA transcripts in iPSC-derived neurons?
4. Fig. 7A-H. The confirmation of signal overlap is shown for 3 paraspeckle proteins (FUS, SFPQ, SMARCC1) but only 1 negative control (PTBP1) is included. Since these proteins are widely distributed throughout the nucleus, several additional widely distributed nuclear RBPs should be analyzed to substantiate the conclusion of selective paraspeckle protein overlap.

Version 1:

Reviewer comments:

Reviewer #1

(Remarks to the Author)

In their revised version, Yap and colleagues have made considerable improvements to the manuscript and addressed all major and most minor concerns raised by the reviewers.

By providing four new supplementary figures (S7-S10) they now confirm in orthogonal experiments (co-localization, co-IP) the RNA association of two select proteins they have identified in their eHyPro (or now HyPro2) approach with HRE+ C9orf72 transcripts. Although this confirmation does not really reflect interactions between endogenous components (RNA and proteins in their respective cell types) it clearly supports their statements.

Furthermore, they have improved figures wherever required (e.g. modified figures 3C, S2B, 6C) and adapted the text (both figure legends and main text in Results and Discussion sections) accordingly.

I congratulate the authors to this now excellent piece of work and improvement of the HyPro method.

Reviewer #3

(Remarks to the Author)

This revision addresses all of my previous concerns and includes several important additional experiments.

Reviewer #1:

Controlling the function of mRNAs requires their assembly with specific sets of proteins. Identifying these proteins and their RNA binding dynamics, both locally and temporally, is of great interest to those seeking to understand cellular RNA function.

Several proximity labeling techniques have been developed in recent years to characterize the interactome ('proxisome') of specific RNAs. Limitations have been the sensitivity (i.e., the bait RNA had to be of minimal abundance) of the approach, the diffusion of the labeling biotin residue (leading to labeling of unwanted bystanders), or the need for in-cell expression of recombinant versions of proximity labeling enzymes, e.g., as part of fusion proteins that target them to the RNA of interest.

The original HyPro technique developed by the Makeyev lab (Mol Cell) has allowed to overcome the second hurdle, but still suffers from the problem of low sensitivity. In this manuscript, the group describes an improved version of HyPro (eHyPro) that overcomes the sensitivity and spreading problems while allowing the use of proximity labeling methods on RNA-associated proteins in unmodified cells.

Major findings:

By carefully studying (and improving) the original HyPro enzyme and labeling procedure, the authors develop an optimized protocol for HyPro-based proximity labeling. This is first demonstrated with in vitro and in cell bulk biotinylation, and then demonstrated on beta-actin RNA transcription sites (TS) and perinucleolar compartments (which are enriched in PNCTR RNA) by detecting biotinylation using fluorescence microscopy. Importantly, the actual proximity labeling experiments for both beta-actin TS and perinucleolar compartments to detect RNA-associated proteins by mass spectrometry also suggest that eHyPro outperforms the original HyPro method. More importantly, they demonstrate for a nuclear (pre-splicing) RNA that eHyPro allows proximity labeling of RNAs with only a few molecules present.

We thank the reviewer for their constructive comments.

Major point of criticism:

Many of the proteins identified as residing in (what the authors call) RNA-proximal compartments or RNA neighborhoods have already been associated with the RNA under study. This makes other proteins identified in these RNA-proximal compartments very likely candidates for modulators of the RNA in question. However, beyond cross-correlation with existing data on these proteins and their interaction in association networks, no additional experimental support for the association of these new factors (such as the paraspeckle markers identified as novel C9orf72 transcript associates) is provided. I suggest to approach this by an orthogonal experimental setup like superresolution microscopy or co-IP.

As suggested, we co-immunoprecipitated formaldehyde-crosslinked RNA-protein complexes as an orthogonal validation approach (see new Figure S10). To demonstrate the type of insights that our proximity labeling technique can provide, we focused on the poorly characterized paraspeckle marker SMARCC1/BAF155 identified by enhanced HyPro-MS as

a cellular neighbor of hexanucleotide repeat expansion-positive (HRE+) *C9orf72* transcripts. We additionally analyzed another HyPro-MS hit, SFPQ, which has been previously shown to interact with HRE+ transcripts in HEK293T overexpression experiments and *C9orf72*-mutant fibroblasts and cerebellar cells (PMID: 30745340), but not in iPSCs.

Given that C9-ALS iPSCs are heterozygous for the HRE+ *C9orf72* allele, and distinguishing between endogenously expressed HRE+ and HRE- transcripts in a biochemical assay is extremely challenging, we examined healthy iPSCs transfected with *C9orf72* minigenes either lacking G4C2 repeats (*miniC9-CS*) or containing 100 G4C2 repeats (*miniC9-100xG4C2*). Following formaldehyde crosslinking and cell lysis, we immunoprecipitated crosslinked RNA-protein complexes using SMARCC1- or SFPQ-specific antibodies, and analyzed the samples by reverse transcription-quantitative PCR (RT-qPCR) using minigene-specific primers.

The *miniC9-100xG4C2* transcripts showed markedly stronger binding to both SMARCC1 and SFPQ compared to *miniC9-CS*. These results, along with additional controls, are presented in the new Figure S10. Of note, we have also confirmed that endogenous HRE+ *C9orf72* transcripts co-localize with SMARCC1 and SFPQ protein densities significantly more often than their HRE- counterparts in two additional C9-ALS iPSC lines, AST2 and M211R2 (new Figures S7-S9 added in response to Reviewer 3).

Minor points:

In line 106, the authors claim that the engineered versions of APEX (APEX, APEX2) have reduced activity compared to the original APX enzyme. These should be checked again. Comparing the k_{cat}/K_m values of APX, engineered APEX, and APEX2 (Martell, Nat. Biotech. 2012; Lam et al., Nat. Meth. 2015), the engineered versions show an increase but no decrease. APX (W41F) by 8.2-fold, mAPX (K14D, E112K) by 1.5-fold, APEX (i.e. mAPX+W41F) by 8.1-fold. This also means that the authors observe an increase in activity from HyPro to eHyPro ok BUT in the original description (Martell, Nat. Biotech) there is no reduction of K14D / E112K compared to wt APX. Perhaps the authors can explain these contrasting observations in the Discussion?

We thank the reviewer for raising this point. We wanted to highlight the finding by Martell *et al.* (Nat. Biotech. 2012; PMID: 23086203) that the K14D and E112K mutations may compromise APX activity under specific conditions, such as in the context of low heme availability. As noted in Martell *et al.*, "We reasoned that the poor activity of mAPX-NES is likely due to less efficient heme incorporation in the cellular context, which is perhaps a result of decreased thermal stability and lower melting temperature associated with monomerization". We have revised Results text on page 4 to clarify our argument, and added a corresponding note in the Discussion on page 12. The overall conclusion from both Martell *et al.* and our current work is that the K14D and E112K mutations can make the enzyme less robust across a broader range of conditions.

Figure 1D: The eHyPro Biotin image shows additional signals that are not detected by FISH in the corresponding partner images. Does this mean that eHyPro can label RNAs that are not detectable by smFISH or that eHyPro can also lead to false positive signals? Regarding the experimental setup with heme and trehalose (e.g., in Figure 3C), why do these

conditions, which are expected to limit diffusion around 'positive' spots, also seem to reduce the number of such putative false-positive signals?

Our interpretation of Figure 1D is that eHyPro (which we renamed to HyPro2, as suggested by Reviewer 2) is simply more active than HyPro, which enhances both specific and nonspecific staining. Regarding Figure 3C, we indeed see consistently lower background staining in the presence of trehalose and hemin compared to the LVB condition. We focused on diffusion-suppressing effects in the original manuscript; however, trehalose also appears to reduce the nonspecific background, possibly by maintaining the enzyme in a properly folded state. We have added a note on this on page 7 and updated Figure 3C and its legend accordingly.

Figure S2B and line 138: The image quality of the image showing a diffraction-limited spot used to quantify the number of individual PNCTR molecules does not appear to be sufficient to verify whether it really reflects a spot, as it appears too blurred.

We have provided a better-quality image in Figure S2B, where the single-molecule PNCTR spots are further away from the main PNC density.

Figure S3A and S3B: What is the difference between labeling efficiency and labeling intensity? The main text and figure legend do not clearly state what these are and why both are shown.

These two statistics were defined in the Figure 1E-F legend. Nonetheless, we have revised the Figure S3 legend to clarify the distinction between labeling efficiency and labeling intensity without referring to the main text.

Figure S6A: The figure shows 3 cells (or 3 nuclei) with a very uneven distribution of signals/dots between them. Is this imbalance a general phenomenon (i.e., is there a large variation in foci between cells of the same type)?

Yes, we frequently observe distinct numbers of HRE+ C9orf72 transcripts across nuclei within the same microscopy field. This variation might be attributed to epigenetic differences at the single-cell level, cell cycle stage-specific effects, stochastic transcriptional bursting, or other mechanisms. A more detailed investigation of this phenomenon is beyond the scope of the current study; however, we have added a brief note highlighting this point in the revised Figure S6A legend.

Figure 6B and lines 310/311: The authors state that "...conditions revealed robust biotinylation of a substantial fraction of C9-Ex-positive RNA signals..." The labeling of the Y-axis suggests values of 0.5 - 1%. Can this really be called 'substantial'?

The "Apparent labeling efficiency" axis represents fractional values, not percentages. We have corrected this error in the revised Figure 6C by removing the "%" symbol. We apologize for the typo.

Figure 7 A to G: It is not clear from the main text and figure legends how exactly the intensity plots were derived. For example, the figure legend states that 'The signal intensity profiles on the right are plotted along the directions indicated by the arrowheads in the main figures'.

If the profile is plotted along a virtual line between the arrowheads, this does not seem to fit the intensity distribution shown.

We have revised Figure 7 legend to clarify our approach: "Signal intensity profiles on the right are plotted along 1.5 μm virtual lines drawn in the direction indicated by the arrowheads in the main images".

Reviewer #2:

This is an interesting methods paper that presents a new approach to detect protein interactomes of selected RNAs. The paper is well written and in step by step modification of their protocol is an improvement on the available techniques. However, it falsely lays claim to novelty regarding paraspeckle component in G4C2 repeat interactome, ignoring/avoiding/diminishing already published findings. This is a major issue and needs to be addressed/corrected.

We apologize if the original manuscript gave the impression of overlooking prior work in this area. We have thoroughly revised the text to more accurately reflect and acknowledge existing findings. Specifically, we now cite the studies by Bajc Česnik *et al.* (2019; PMID: 30745340) and Malnar and Rogelj (2021; PMID: 33495278) in the Results and reference them again in the Discussion. Moreover, we have expanded the study by adding new data (Figures S7-S10), shifting the focus toward SMARCC1/BAF155, a previously uncharacterized interactor of mutant C9orf72 transcripts, which is known to interact with paraspeckles (PMID: 25831520). In contrast, SFPQ is now used primarily as a positive control.

L35. 'Uncovered' needs to be changed to 'confirmed'. Using 'uncovered' is misleading as the technique only confirms an already published paraspeckle-like properties of the G4C2 repeat (Bajc Cesnik *et al.*, J Cell Biol 2019) and further confirmed by Malnar *et al.*, J Cell Biol 2021.

We have replaced "uncovered" with the more neutral term "identified" to moderate the implication of novelty. We believe the revised manuscript now clearly distinguishes between previously known interactors – such as SFPQ and FUS – and less well-characterized or new interactors, such as SMARCC1.

L37. 'reveal' is also an overclaim, because the findings are known.

We have revised the entire sentence to avoid overstatement (see revised Abstract). It now reads: "These findings highlight early RNA processing and localization defects in ALS that might contribute to this late-onset neurodegenerative disorder".

L104. Consider renaming eHyPro as it is odd to name something as enhanced, while the process is actually reverting the mutations introduced in APEX2 and removing a T7 tag.

As suggested, we have renamed the modified enzyme from "eHyPro" to the more neutral "HyPro2".

L335. 'gratifyingly' is not a scientific term. In the following lines it should be mentioned that Bajc Cesnik paper needs mentioning.

"Gratifyingly" has been deleted and Bajc Česnik et al. (2019) and Malnar and Rogelj (2021) are now referenced in the following lines.

L343-351. These experiments are just a repeat of what has already been shown in Bajc Cesnik for patient fibroblasts and in case of SFPQ, patient brains. This should be mentioned.

We have substantially revised this section, citing Bajc Česnik et al. (2019) and Malnar and Rogelj (2021) and adding several new experiments (Figures S7-S10), which shift the focus of this part of our study from SFPQ and FUS to SMARCC1.

L448. The only mention of the Bajc Česnik paper is here and it is very vague. It should be mentioned that the study differentiated cells are C9orf72 patient fibroblasts. It is completely omitted that there is data on colocalization in C9orf72 brains. Also the manuscript should mention that SFPQ regulates G4C2 foci formation (Malnar et al. J Cell Sci 2021).

We have specified the cell types investigated by Bajc Česnik et al. (2019) in the revised Results (page 11) and Discussion (page 14). The role of SFPQ in the formation of G4C2 RNA foci uncovered by Malnar and Rogelj (2021) is now mentioned in the Results (page 11).

L453-456. The statement is misleading. What is the proof that the clusters of paraspeckle proteins are small? Also, G4C2 foci are NEAT1 independent (See Bajc Cesnik, J Cell Biol 2019), therefore there is no reason why there should be some mystery in the interaction of G4C2 with paraspeckle proteins in the situation where there are no paraspeckles.

As suggested, we have removed the reference to "small clusters" in the Discussion (pages 14-15), as we agree there is no direct evidence supporting a size-based characterization in this context. Additionally, we have toned down the paragraph that mentioned the absence of NEAT1_2 expression in iPSCs in the original version (page 11; paragraph "To find out if mutant C9orf72 transcripts interact with paraspeckle proteins in pluripotent stem cells...").

We thank the reviewer for their insightful comments, which have helped us present a more nuanced and accurate interpretation of the C9orf72/paraspeckle relationship in the revised manuscript.

Reviewer #3:

Multiple proximity labeling strategies have been described that are designed to identify the proteins bound to specific RNAs. Here, Yap et al report a modification of their hybridization-based proximity (HyPro) labeling protocol, or enhanced (e)HyPro, to optimize the identification of proteins interacting with single RNA molecules. Modification of the HyPro APEX2 domain to increase peroxidase activity and DIG binding resulted in eHyPro labeling of ACTB transcription sites using both exon and intron probes. Since biotin label diffusion was observed with eHyPro, they next limit this diffusion using 50% trehalose while also demonstrating hemin preincubation (5 μ M) enhances proximity labeling. PC analysis of

eHyPro MS results showed distinct clustering of control, PNCTR and ACTB proteome profiles with multiple RBPs and nucleoli-associated proteins identified in the PNCTR proteome, including integrator (INTS4) and exosome (EXOSC10) proteins, together with a similar analysis of ACTB transcription sites that also highlighted several NUP proteins which led the authors finding a nuclear periphery bias of these sites. Finally, they identify the C9orf72 RNA interactome in control vs C9-ALS iPSCs and identify interactions between single RNA foci, paraspeckle and splicing proteins. Overall, this is a careful and well-written study that reports a technical advancement in the proximity labeling field, but I have a few concerns noted below.

1. Abstract, In 28. The authors describe eHyPro for proteomic mapping of RNA microcompartments (<10 RNA molecules) but designate their prime example, C9orf72 mutant iPSCs with expanded G4C2 repeats, as 'genetically unperturbed cells'. This is confusing when referring to a genetic mutant and it might be clearer to simply state that the iPSCs are only expressing endogenous RNA transcripts.

We agree that referring to iPSCs carrying a disease-associated repeat expansion as "genetically unperturbed" was potentially confusing. The revised sentence now reads: "We present an enhanced hybridization-proximity labeling (HyPro) technology for mapping the proteomes of endogenously expressed RNA microcompartments".

2. Results, In 333. Fig. S6E does not exist – I assume the authors were referring to S6C.

We apologize for the typo. The correct reference should be to Fig. S5E – fixed in the revised manuscript.

3. Fig. 6. This figure describes a proteomic composition analysis based on only single male healthy and c9-ALS iPSCs. First, these results should be confirmed using iPSCs from multiple donors since these are readily available. Second, ALS affects primarily motor and cortical neurons, and thus a more disease relevant analysis would be to identify the protein composition associated with C9 RNA transcripts in iPSC-derived neurons?

We thank the reviewer for raising these important points. To address the issue of donor representation, we have now validated the interactions between mutant C9orf72 transcripts and both a previously established partner (SFPQ) and a newly identified one (SMARCC1) in two additional C9-ALS iPSC lines. The new data, presented in Figures S8 and S9, confirm the findings obtained in the original version of the manuscript. Regarding disease relevance, we fully agree that characterizing RNA-protein interactions in motor neurons would be highly informative. However, such an analysis should ideally involve a systematic, time-resolved investigation of transcript dynamics and ribonucleoprotein remodeling during the motor neuron differentiation process – a direction that extends beyond the scope of our methods-focused manuscript. We are planning to perform such time-resolved HyPro-MS experiments as part of our future studies.

4. Fig. 7A-H. The confirmation of signal overlap is shown for 3 paraspeckle proteins (FUS, SFPQ, SMARCC1) but only 1 negative control (PTBP1) is included. Since these proteins are widely distributed throughout the nucleus, several additional widely distributed nuclear RBPs should be analyzed to substantiate the conclusion of selective paraspeckle protein overlap.

As suggested, we have included OCT4 as an additional negative control in the new Figures S8 and S9. OCT4 is highly expressed in iPSCs and is known to associate with many chromatin regions throughout the nucleus, making it a suitable control for testing the specificity of paraspeckle protein co-localization with C9orf72 transcripts.